# ADAPTIVE KERNEL SELECTION FOR STEIN VARIATIONAL GRADIENT DESCENT

## ABSTRACT

A central challenge in Bayesian inference is efficiently approximating posterior distributions. Stein Variational Gradient Descent (SVGD) is a popular variational inference method which transports a set of particles to approximate a target distribution. The SVGD dynamics are governed by a reproducing kernel Hilbert space (RKHS) and are highly sensitive to the choice of the kernel function, which directly influences both convergence and approximation quality. The commonly used median heuristic offers a simple approach for setting kernel bandwidths but lacks flexibility and often performs poorly, particularly in high-dimensional settings. In this work, we propose an alternative strategy for adaptively choosing kernel parameters over an abstract family of kernels. Recent convergence analyses based on the kernelized Stein discrepancy (KSD) suggest that optimizing the kernel parameters by maximizing the KSD can improve performance. Building on this insight, we introduce Adaptive SVGD (Ad-SVGD), a method that alternates between updating the particles via SVGD and adaptively tuning kernel bandwidths through gradient ascent on the KSD. We provide a simplified theoretical analysis that extends existing results on minimizing the KSD for fixed kernels to our adaptive setting, showing convergence properties for the maximal KSD over our kernel class. Our empirical results further support this intuition: Ad-SVGD consistently outperforms standard heuristics in a variety of tasks.

## 1 INTRODUCTION

Stein Variational Gradient Descent (SVGD) (Liu & Wang, 2016) is a deterministic particle-based method for approximate Bayesian inference that has emerged as a popular alternative to traditional Markov Chain Monte Carlo (MCMC) methods. SVGD evolves a set of particles using update directions derived from the functional gradient of the Kullback-Leibler (KL) divergence to the target distribution, with updates constrained to lie within the unit ball of a reproducing kernel Hilbert space (RKHS). A critical limitation of SVGD is its sensitivity to kernel choice, which significantly influences the algorithm's performance (Duncan et al., 2023; Nüsken & Renger, 2023). Additionally, the resulting particle approximation commonly underestimates the posterior variance (Ba et al., 2022). These observations have led to the widely held belief that SVGD in general fails to perform well as the dimension of the underlying state space increases. In this work, we challenge this belief by introducing an adaptive mechanism for selecting kernel parameters that dynamically tunes the kernel during inference by maximizing the kernelized Stein discrepancy (KSD), enabling more effective transport in complex and high-dimensional spaces.

### 1.1 RELATED WORK.

Since its introduction (Liu & Wang, 2016), SVGD has become a widely used tool for approximate Bayesian inference in a range of machine learning applications (Liu et al., 2017; Messaoud et al., 2024; Pu et al., 2017; Kassab & Simeone, 2022). Recent work has made substantial progress in understanding the theoretical underpinnings of SVGD. Mean-field convergence has been analyzed in both continuous-time (Lu et al., 2019; Duncan et al., 2023; Chewi et al., 2020) and discrete-time (Korba et al., 2020; Salim et al., 2022) settings, while finite-particle convergence rates have been established under various assumptions (Balasubramanian et al., 2025; Shi & Mackey, 2023). The SVGD

dynamics have also been connected to gradient flows on probability distribution spaces (Liu, 2017; Duncan et al., 2023), drawing analogy to Jordan et al. (1998).

The performance of SVGD depends critically on the choice of the kernel function, as it determines the interaction between particles and the overall convergence of the method (see also Figures 1 and 8). Convergence results typically refer to mean-field convergence with respect to KSD, whose relation to weak convergence depends on the selected kernel (Gorham & Mackey, 2017). The commonly used median heuristic (Gretton et al., 2012) provides a simple implementation but lacks theoretical justification and is known to degrade in performance as the dimensionality of the problem increases (Ba et al., 2022; Zhuo et al., 2018; Wang et al., 2018). Recent work has developed tools to mitigate performance degradation in high-dimensional settings (Detommaso et al., 2018; Wang et al., 2019; Gong et al., 2021; Liu et al., 2022). The algorithm has also been modified through the use of neural networks to learn the update direction (di Langosco et al., 2022; Zhao et al., 2023).

The approach most closely related to our work is Ai et al. (2023), which introduces a mixture-of-kernels framework. Their method defines a KSD for a weighted linear combination of kernels and learns the kernel weights by maximizing this multiple-kernel KSD. However, their approach is limited to finite kernel bases and does not explore continuous parameter optimization as proposed in our work.

## 1.2 CONTRIBUTIONS.

In this work, we address the fundamental research question of whether SVGD, without any architectural or algorithmic enhancements, can achieve competitive performance when equipped with a principled and adaptively chosen kernel. By isolating kernel selection from other modifications found in SVGD variants, we aim to rigorously understand the extent to which kernel choice alone governs the effectiveness of SVGD and how far adaptive kernel learning can push the capabilities of the original method.

Our main contributions are as follows.

(i) **Adaptive Kernel Selection Method.** We propose a novel method that dynamically updates the kernel parameters by maximizing the KSD during SVGD inference. In contrast to the commonly used median heuristic, which relies on a single scalar bandwidth, our approach allows for the optimization of multiple continuous kernel parameters, enabling greater flexibility and adaptivity during SVGD updates.

(ii) **Theoretical Analysis.** We provide theoretical motivation by analyzing our algorithm in the discrete-time mean-field setting, extending existing convergence results for SVGD with fixed kernels. Specifically, we show that the supremum of the KSD over a parameterized kernel class converges to zero as the particle distribution approaches the target. Assuming a Stein logarithmic Sobolev inequality we further derive iteration complexity in the mean-field limit.

(iii) **Empirical Validation.** Through numerical experiments, we demonstrate that our adaptive kernel selection consistently outperforms the median heuristic and helps alleviate variance collapse.

## 2 MATHEMATICAL BACKGROUND

We begin by considering a symmetric positive definite kernel $k : \mathbb{R}^d \times \mathbb{R}^d \to \mathbb{R}$ and its associated RKHS $\mathcal{H}_0$. We define $\mathcal{H}$ as the $d$-fold Cartesian product of $\mathcal{H}_0$ equipped with the inner product $\langle f, g \rangle_{\mathcal{H}} = \sum_{i=1}^d \langle f_i, g_i \rangle_{\mathcal{H}_0}$ and the canonical feature map $\Phi_k(x) = k(\cdot, x) \in \mathcal{H}_0$. Moreover, we denote by $x \cdot y$ the Euclidean inner product and $\nabla \cdot$ the divergence operator. The space of probability measures on $(\mathbb{R}^d, \mathcal{B}(\mathbb{R}^d))$ is denoted by $\mathcal{P}(\mathbb{R}^d)$ and $\mathcal{P}_p(\mathbb{R}^d)$ denotes the subspace of measures with finite $p$-th moment. For $\mu, \nu \in \mathcal{P}_p(\mathbb{R}^d)$, we define the Wasserstein $p$-distance

$$\mathcal{W}_p(\mu, \nu) = \inf_{\gamma \in \Gamma(\mu, \nu)} \Big( \int_{\mathbb{R}^d \times \mathbb{R}^d} \|x - y\|_2^p \mathrm{d}\gamma(x, y) \Big)^{1/p},$$

where $\Gamma(\mu, \nu)$ is the set of couplings of $\mu$ and $\nu$, i.e. the set of probability measures on $\mathbb{R}^d \times \mathbb{R}^d$ with marginals $\mu$ and $\nu$.

## 2.1 INTEGRAL PROBABILITY METRICS AND KERNELIZED STEIN DISCREPANCY.

*Integral probability metrics* (IPMs) (Müller, 1997) are a way to quantify the distance between two measures by considering the maximum deviation of integrals over a class of test functions. To make this approach feasible for measuring the distance of a sample to an intractable target distribution, Stein's method (Stein, 1972) can be used to construct test functions which have zero mean w.r.t. the target. Indeed, for the operator $\mathcal{S}_\pi f := \nabla \log \pi \cdot f + \nabla \cdot f$ and a suitable choice of kernel, we have $\int_{\mathbb{R}^d} \mathcal{S}_\pi f(x) \mathrm{d}\pi(x) = 0$ for all $f \in \mathcal{H}$ (Chwialkowski et al., 2016; Liu et al., 2016). They defined the *kernelized Stein discrepancy* (KSD) as

$$\mathrm{KSD}(\mu|\pi) := \sup_{f \in B(\mathcal{H})} \left| \int_{\mathbb{R}^d} \mathcal{S}_\pi f \mathrm{d}\mu \right|, \tag{1}$$

where $B(\mathcal{H})$ denotes the unit ball in $\mathcal{H}$. This optimization problem is solved by $f^* = \frac{\psi}{\|\psi\|_{\mathcal{H}}}$ with $\psi = \int_{\mathbb{R}^d} \mathcal{A}_\pi^k \mathrm{d}\mu$, where $\mathcal{A}_\pi^k(x) = \nabla \log \pi(x) \Phi_k(x) + \nabla \Phi_k(x) \in \mathcal{H}$, and $\Phi_k$ is the feature map associated with the kernel $k$. As a result, the supremum evaluates to the RKHS norm of $\psi$, giving the equivalent characterization $\mathrm{KSD}(\mu|\pi) = \|\psi\|_{\mathcal{H}}$.

## 2.2 STEIN VARIATIONAL GRADIENT DESCENT.

Given a target distribution $\pi$ and reference distribution $\mu_0$, SVGD transforms $\mu_0$ into an approximation of $\pi$ by choosing $\mu_{n+1} := T_\sharp \mu_n$, where $T_\sharp \cdot$ is the push-forward operator for the map $T(x) = x + \gamma \psi^{\mu_n}(x)$, with the vector field $\psi^\mu = \int_{\mathbb{R}^d} \mathcal{A}_\pi^k \mathrm{d}\mu$ for $\mu \in \mathcal{P}(\mathbb{R}^d)$ being the direction of steepest descent. This is motivated by the fact that the solution of Equation (1) implies that $\frac{\psi^{\mu_n}}{\|\psi^{\mu_n}\|_{\mathcal{H}}}$ is the minimizer of $\frac{\mathrm{d}}{\mathrm{d}\gamma} \mathrm{KL}(T_\sharp \mu_n \| \pi)\big|_{\gamma=0}$ in the unit ball of $\mathcal{H}$ (cf. Liu & Wang, 2016, Theorem 3.1), where $\mathrm{KL}(\cdot \| \cdot)$ denotes the KL-divergence. In particular, for this choice we have

$$\frac{\mathrm{d}}{\mathrm{d}\gamma} \mathrm{KL}\Big( \big( \mathrm{Id} + \gamma \psi^{\mu_n} \big)_\sharp \mu_n \,\big\|\, \pi \Big)\Big|_{\gamma=0} = -\mathrm{KSD}^2(\mu_n|\pi). \tag{2}$$

Iteratively applying the maps $T$ generated in this way to a particle set $\{X_0^i\}_{i=1}^M$ sampled from $\mu_0$ leads to the following particle updates:

$$X_{n+1}^i = X_n^i + \frac{\gamma}{M} \sum_{j=1}^M k(X_n^i, X_n^j) \nabla \log \pi(X_n^j) + \nabla_{X_n^j} k(X_n^i, X_n^j). \tag{3}$$

## 3 ADAPTIVE KERNEL SELECTION FOR SVGD

Since proving convergence of SVGD with respect to the KL divergence is challenging and requires restrictive assumptions, recent work has shifted attention to analyzing convergence in terms of the kernelized Stein discrepancy (KSD) (Korba et al., 2020; Shi & Mackey, 2023; Salim et al., 2022). However, minimizing the KSD alone does not guarantee weak convergence: the sequence of mean-field measures may fail to be tight, which is a necessary condition for weak convergence of probability measures in Polish spaces. Moreover, as seen from the optimization problem in Equation (1), the RKHS structure (and therefore the choice of kernel) directly determines how well convergence in KSD translates into weak convergence (Gorham & Mackey, 2017).

To address this issue and strengthen convergence guarantees, we formulate our adaptive variant of SVGD using a parameterized family of kernels $\{k_\theta \mid \theta \in \Theta\}$. For each $\theta \in \Theta$, we denote by $\mathrm{KSD}_\theta$ the corresponding *kernelized Stein discrepancy*, by $\psi_\theta^\mu$ the optimal update direction, and by $\Phi_\theta$ the associated feature map. This explicit parameterization allows us to adaptively select kernels during optimization. In doing so, we account for the kernel's influence on convergence properties and mitigate the limitations of relying on a fixed kernel choice.

Our approach builds on the idea that it is advantageous to select a kernel that maximizes the KSD between the empirical particle distribution and the target distribution. The intuition is straightforward: the instantaneous decrease in KL divergence under SVGD is proportional to the squared KSD at the current particle measure (see Equations (2) and (7)). Thus, at any given iteration, choosing

the kernel that yields the largest KSD corresponds to maximizing the rate of KL decrease. This perspective is reinforced by geometric analyses of SVGD as a gradient flow, which show that kernels inducing larger KSD values yield more favorable convergence properties when comparing the associated RKHSs (Nüsken & Renger, 2023; Duncan et al., 2023).

While vanilla SVGD usually aims to guarantee convergence of KSD for a fixed kernel, our proposed algorithm targets the worst-case KSD within the kernel class. As opposed to the median heuristic used by Liu & Wang (2016), the approach is applicable to any class of parameterized kernels: At each step of the algorithm, we try to find an optimal parameter

$$\theta_n \in \underset{\theta \in \Theta}{\operatorname{argmax}} \operatorname{KSD}_\theta(\mu_n | \pi) .$$

We implement this idea by adjusting the kernel parameter using one or possibly more gradient ascent steps with a step size $s > 0$ for KSD before executing the particle update and also allow the possibility of not updating the parameter at every step of SVGD to decrease runtime (see Algorithm 1). To enable this option, we introduce a user-specified decision variable $\mathrm{paramupdate}$. It is worth noting that the base SVGD step for transporting the particles can be replaced by more advanced variants (e.g., adaptive step-size schedules, line searches, momentum, or second-order/preconditioned updates) without modifying our kernel-selection mechanism. Similarly, we may replace the gradient ascent on the kernel parameter with alternative optimization schemes. Our implementation of the kernel update is based on the formula (Chwialkowski et al., 2016; Liu et al., 2016)

$$\operatorname{KSD}^2(\mu | \pi) = \int_{\mathbb{R}^d \times \mathbb{R}^d} u_\pi^k \, \mathrm{d}(\mu \otimes \mu) \tag{4}$$

with

$$\begin{aligned}
u_\pi^k(x, y) = {} & k(x, y) \nabla \log \pi(x) \cdot \nabla \log \pi(y) + \nabla \log \pi(y) \cdot \nabla_x k(x, y) \\
& + \nabla \log \pi(x) \cdot \nabla_y k(x, y) + \operatorname{trace}(\nabla_x \nabla_y k(x, y)),
\end{aligned} \tag{5}$$

from which the necessary gradient can be computed directly. It is important to note two comments regarding the additional computational cost of Ad-SVGD compared to vanilla SVGD:

- The gradient ascent steps for the kernel parameter use the same gradients $\nabla \log \pi(X_n^i)$ as the corresponding SVGD step and Ad-SVGD does therefore not require additional gradient evaluations of $\log \pi$.

- The associated computational cost can be further reduced by relying on a small number of update steps $\mathrm{nstepstheta}$ for the kernel parameter or optionally updating the kernel parameter only after multiple SVGD steps. When the number of particles $M$ is large, one may use subsampled particles to empirically approximate the KSD.

---

**Algorithm 1:** Ad-SVGD

---

**Input:** Initial particle set $\{X_0^i \in \mathbb{R}^d \mid i = 1, \dots, M\}$, kernel class $\{k_\theta \mid \theta \in \Theta\}$, initial kernel parameter $\theta_{-1} \in \Theta$, step sizes $\gamma, s > 0$, number of steps $\mathrm{nsteps}, \mathrm{nstepstheta} \in \mathbb{N}$
**Output:** Final particle set $\{X_{\mathrm{nsteps}}^i \in \mathbb{R}^d \mid i = 1, \dots, M\}$

---

**for** $n = 0$ **to** $\mathrm{nsteps} - 1$ **do**
    Decide $\mathrm{paramupdate} \in \{\mathrm{True}, \mathrm{False}\}$;
    **if** $\mathrm{paramupdate}$ **then**
        $\theta_n^0 \leftarrow \theta_{n-1}$;
        **for** $\ell = 0$ **to** $\mathrm{nstepstheta} - 1$ **do**
            $\theta_n^{\ell+1} \leftarrow \theta_n^\ell + s \, \nabla_{\theta_n^\ell} \operatorname{KSD}_{\theta_n^\ell}^2 \left( \frac{1}{M} \sum_{i=1}^M \delta_{X_n^i} \middle| \pi \right)$;
        **end**
        $\theta_n \leftarrow \theta_n^{\mathrm{nstepstheta}}$;
    **else**
        $\theta_n \leftarrow \theta_{n-1}$;
    **end**
    **for** $i = 1$ **to** $M$ **do**
        $X_{n+1}^i \leftarrow X_n^i + \frac{\gamma}{M} \sum_{j=1}^M k_{\theta_n}(X_n^i, X_n^j) \nabla \log \pi(X_n^j) + \nabla_{X_n^j} k_{\theta_n}(X_n^i, X_n^j)$;
    **end**
**end**

---

## 4 CONVERGENCE ANALYSIS

We consider a target measure with Lebesgue density of the form $\pi(x) \propto \exp(-V)$ for a potential $V : \mathbb{R}^d \to \mathbb{R}$. To better motivate our proposed Ad-SVGD, we will demonstrate how to extend recent results on the convergence of SVGD in the sense of KSD. To be more precise, we will extend the convergence analysis conducted in Salim et al. (2022) under the Talagrand's inequality which holds under mild assumptions on the target distribution and is weaker than the commonly employed logarithmic Sobolev inequality. We make the same assumptions as in Salim et al. (2022) uniform over all kernel parameters.

**Assumption 1.** *We assume that $V \in C^2$ such that $\int_{\mathbb{R}^d} \exp(-V(x)) \, dx < \infty$ and that the Hessian $H$ of $V$ is uniformly bounded w.r.t. the operator norm, i.e. there exists $L \geq 0$ such that $\|H(x)\|_{\mathrm{op}} \leq L$ for all $x \in \mathbb{R}^d$.*

**Assumption 2.** *We assume that $\pi \in \mathcal{P}_1(\mathbb{R}^d)$ satisfies Talagrand's inequality $T1$, which means that there exists $\lambda > 0$ such that*

$$\mathcal{W}_1(\mu, \pi) \leq \sqrt{2 \, \mathrm{KL}(\mu \,\|\, \pi)/\lambda}$$

*for all $\mu \in \mathcal{P}_1(\mathbb{R}^d)$.*

**Assumption 3.** *We assume there exists $B > 0$ such that $\|\Phi_\theta(x)\|_{\mathcal{H}_0} \leq B$ for all $x \in \mathbb{R}^d$, $\theta \in \Theta$, and that $\nabla \Phi_\theta$ is continuous with $\|\nabla \Phi_\theta(x)\|_{\mathcal{H}}^2 = \sum_{i=1}^d \|\partial_i \Phi_\theta(x)\|_{\mathcal{H}_0}^2 \leq B^2$ for all $x \in \mathbb{R}^d$, $\theta \in \Theta$.*

### 4.1 CONVERGENCE ANALYSIS OF SVGD UNDER TALAGRAND'S INEQUALITY

In the following, we denote

$$\mathcal{F}(\mu) := \mathrm{KL}(\mu \,\|\, \pi).$$

and make use of the following fundamental inequality. Given a fixed kernel parameter $\theta \in \Theta$ such that Assumption 3 is satisfied, define the pushforward measure

$$\tilde{\mu} = (I + \gamma g) \# \mu$$

for arbitrary $g \in \mathcal{H}$. Under Assumptions 1 and 3, let $\gamma, B > 0$, $\alpha > 1$ and $g \in \mathcal{H}_\theta$ such that $\gamma \|g\|_{\mathcal{H}_\theta} \leq \frac{\alpha-1}{\alpha B}$. Then, according to Proposition 3.1 in Salim et al. (2022), it holds

$$\mathcal{F}(\tilde{\mu}) \leq \mathcal{F}(\mu) + \gamma \langle \psi_\theta^\mu, g \rangle_{\mathcal{H}_\theta} + \frac{\gamma^2 K}{2} \|g\|_{\mathcal{H}_\theta} \tag{6}$$

with $K = (\alpha^2 + L)B$. For the iterates (3) with a fixed kernel parameter $\theta \in \Theta$ one can then derive the descent condition (Salim et al., 2022, Theorem 3.2),

$$\mathcal{F}(\mu_{n+1}) \leq \mathcal{F}(\mu_n) - \gamma(1 - \frac{\gamma B(\alpha^2 + L)}{2}) \mathrm{KSD}_\theta^2(\mu_n | \pi), \tag{7}$$

provided that

$$\gamma \leq (\alpha - 1) \Big( \alpha B \big( 1 + \|\nabla V(0)\| + L \int_{\mathbb{R}^d} \|x\| \, d\pi(x) + L\sqrt{\frac{2\mathcal{F}(\mu_0)}{\lambda}} \big) \Big)^{-1}. \tag{8}$$

The key aspect to verify (7) is the verification of $\gamma \|\psi_\theta^{\mu_n}\|_{\mathcal{H}_\theta} \leq \frac{\alpha-1}{\alpha B}$ using Talagrand's inequality, Assumption 2, which allows to apply (6) for $g = \psi_\theta^{\mu_n}$. Condition (7) can then be used to argue that

$$\lim_{n \to \infty} \mathrm{KSD}_\theta^2(\mu_n | \pi) = 0,$$

since $\sum_{n=0}^\infty \mathrm{KSD}_\theta^2(\mu_n | \pi) \leq c_\gamma^{-1} \mathcal{F}(\mu_0)$ for $c_\gamma = \gamma(1 - \frac{\gamma B(\alpha^2 + L)}{2}) > 0$.

When introducing an adaptive kernel parameter choice $(\theta_n)$, the inequality (7) changes to

$$\mathcal{F}(\mu_{n+1}) \leq \mathcal{F}(\mu_n) - \gamma(1 - \frac{\gamma B(\alpha^2 + L)}{2}) \mathrm{KSD}_{\theta_n}^2(\mu_n | \pi).$$

### 4.2 Convergence analysis of Ad-SVGD.

Suppose the adaptive SVGD iteration can be written in the following simplified form

$$\mu_{n+1} = (I + \gamma \psi_{\theta_n}^{\mu n})_\sharp \mu_n,$$
$$\theta_n \in \underset{\theta \in \Theta}{\operatorname{argmax}} \operatorname{KSD}_\theta^2(\mu_n | \pi), \tag{9}$$

meaning that

$$\|\psi_{\theta_n}^{\mu_n}\|_{\mathcal{H}_{\theta_n}} = \max_{\theta \in \Theta} \operatorname{KSD}_\theta(\mu_n | \pi).$$

We emphasize that this formulation is possible when the maximization of the KSD with respect to the kernel parameter has a (unique) closed-form solution. This is actually the case in the multiple-kernel SVGD framework Ai et al. (2023), which parameterizes the kernel as a convex combination of base kernels

$$k_\theta(x, y) = \sum_{i=1}^N \theta_i k_i(x, y) \quad \text{with} \sum_{i=1}^N \theta_i = 1.$$

For the general setting, the kernel parameter update $\theta_n$ in (9) needs to be approximated. Note that in the general setup we assume that $\operatorname{argmax}_{\theta \in \Theta} \operatorname{KSD}_\theta^2(\mu_n | \pi)) \neq \emptyset$ for all $n \in \mathbb{N}$.

The following descent condition is a direct consequence of Theorem 3.2 in Salim et al. (2022):

**Lemma 1.** *Suppose that Assumptions 1-3 are satisfied. For any $\alpha > 1$ with*

$$\gamma \leq (\alpha - 1)\Big(\alpha B\big(1 + \|\nabla V(0)\| + L \int_{\mathbb{R}^d} \|x\| \, d\pi(x) + L\sqrt{\frac{2\mathcal{F}(\mu_0)}{\lambda}}\big)\Big)^{-1},$$

*there exists $c_\gamma > 0$ such that for all $n \in \mathbb{N}$*

$$\mathcal{F}(\mu_{n+1}) \leq \mathcal{F}(\mu_n) - c_\gamma \max_{\theta \in \Theta} \operatorname{KSD}_\theta^2(\mu_n | \pi),$$

*where $(\mu_n)_{n \in \mathbb{N}}$ is generated by (9).*

**Corollary 2.** *Under the same assumptions as in Lemma 1 it holds that $\lim_{n \to \infty} \max_{\theta \in \Theta} \operatorname{KSD}_\theta^2(\mu_n | \pi) = 0$.*

The situation changes when we assume that we can only approximately solve the maximization task of the KSD with respect to the kernel parameter. Suppose that $\max_{\theta \in \Theta} \operatorname{KSD}_\theta(\mu | \pi) < \infty$ for any $\mu \in \mathcal{P}(\mathbb{R}^d)$ and consider the alternating scheme

$$\theta_n = \Psi_n(\theta_{n-1}, \mu_n),$$
$$\mu_{n+1} = (\operatorname{Id} + \gamma \psi_{\theta_n}^{\mu_n})_\sharp \mu_n \tag{10}$$

for some sequence of iterative update rules $\Psi_n : \Theta \times \mathcal{P}(\mathbb{R}^d) \to \Theta$ with the goal to maximize $\operatorname{KSD}_\theta(\mu_n | \pi)$. Specifically, in Algorithm 1 $\Psi_n$ corresponds to the gradient ascent update for the KSD. However, as mentioned above, this update could be replaced by other suitable iterative optimization schemes. Our required assumption on the update rule is the following convergence behavior.

**Assumption 4.** *We assume that there exists a sequence $(\varepsilon_n)_{n \in \mathbb{N}}$ such that $\sum_{n=0}^\infty \varepsilon_n < \infty$ and*

$$\max_{\theta \in \Theta} \operatorname{KSD}_\theta^2(\mu_n | \pi) - \operatorname{KSD}_{\theta_n}^2(\mu_n | \pi) \leq \varepsilon_n \qquad \text{for all } n \in \mathbb{N}.$$

Using this assumption, we can make the following convergence guarantee.

**Theorem 3.** *Suppose that Assumptions 1-3 are satisfied. Under Assumption 4, for any $\alpha > 1$ with*

$$\gamma \leq (\alpha - 1)\Big(\alpha B\big(1 + \|\nabla V(0)\| + L \int_{\mathbb{R}^d} \|x\| \, d\pi(x) + L\sqrt{\frac{2\mathcal{F}(\mu_0)}{\lambda}}\big)\Big)^{-1},$$

*it holds that $\lim_{n \to \infty} \max_{\theta \in \Theta} \operatorname{KSD}_\theta(\mu_n | \pi) = 0$, where $(\mu_n)_{n \in \mathbb{N}}$ is generated by (10).*

*Proof.* Using Theorem 3.2 in Salim et al. (2022) we obtain

$$
\begin{aligned}
\mathcal{F}(\mu_{n+1}) &\leq \mathcal{F}(\mu_n) - c_\gamma \, \mathrm{KSD}^2_{\theta_n}(\mu_n|\pi) \\
&= \mathcal{F}(\mu_n) - c_\gamma \max_{\theta \in \Theta} \mathrm{KSD}^2_\theta(\mu_n|\pi) + c_\gamma \big( \max_{\theta \in \Theta} \mathrm{KSD}^2_\theta(\mu_n|\pi) - \mathrm{KSD}^2_{\theta_n}(\mu_n|\pi) \big) \\
&\leq \mathcal{F}(\mu_n) - c_\gamma \max_{\theta \in \Theta} \mathrm{KSD}^2_\theta(\mu_n|\pi) + c_\gamma \varepsilon_n \, .
\end{aligned}
$$

Iterating this inequality over $n$ yields

$$
c_\gamma \sum_{n=0}^\infty \max_{\theta \in \Theta} \mathrm{KSD}^2_\theta(\mu_n|\pi) \leq \mathcal{F}(\mu_0) + c_\gamma \sum_{n=0}^\infty \varepsilon_n < \infty \, .
$$

$\square$

Under the so-called Stein logarithmic Sobolev inequality, we can further extend the convergence of Ad-SVGD to convergence in the KL divergence. Under specific scenarios for $\varepsilon_n$ we derive iteration complexity bounds for the mean-field limiting system (10). Moreover, this setting allows us to explicitly describe the bias of Ad-SVGD when the KSD maximization is only solved up to a fixed $\bar{\varepsilon}$-accuracy in each iteration. The details can be found in Section B. Finally, we emphasize that we require only one kernel in our kernel class $\{k_\theta\}$ to satisfy the Stein logarithmic Sobolev inequality which yields a promising flexibility to verify this condition.

## 5 NUMERICAL EXPERIMENTS

In the following section, we evaluate Ad-SVGD on two numerical experiments: a one-dimensional Gaussian mixture model and a linear inverse problem governed by an ordinary differential equation. Additional toy experiments examining the effect of increasing dimensionality are provided in Section A. Moreover, in Section 5.4 we demonstrate the advantages of Ad-SVGD over SVGD with the median heuristic on a Bayesian logistic regression model, which was also used in prior SVGD literature.

### 5.1 KERNEL PARAMETERIZATION

SVGD is most commonly used with kernels of the form

$$
k_h(x,y) = \exp\left( -\frac{\|x - y\|_p^p}{h} \right) ,
$$

where $\|\cdot\|_p$ denotes the $p$-norm on $\mathbb{R}^d$ (e.g. Liu & Wang, 2016; Ba et al., 2022; Duncan et al., 2023). We will focus on selection strategies for the parameter $h$, which is known as the kernel *bandwidth*. The commonly used heuristic sets $h = \frac{\mathrm{med}^p}{\log(M-1)}$, where med denotes the current median distance between the particles. This choice is motivated by the goal of balancing the two terms in the SVGD update (3) (Liu & Wang, 2016).

To take advantage of the flexibility of our adaptive method, we use product kernels of the form

$$
k_h(x,y) = \prod_{i=1}^d \exp\left( -\frac{|x_i - y_i|^p}{h_i} \right)
$$

with parameter $h = (h_1, \ldots, h_d)$, i.e. we allow for dimension-dependent bandwidths. The derivatives necessary to apply Algorithm 1 with these kernels (see Equations (4) and (5)) can easily be calculated. We also tested using an adjusted version of the median heuristic with these kernels taking a naive median for each dimension. However, this approach did not produce good results and suffered from a variance collapse.

Beyond our current experiments, the flexibility of Ad-SVGD allows for more scalable kernel families in higher-dimensional settings. In particular, spectral or low-rank kernel parametrizations can adapt the dominant directions of variability without incurring the exponential cost of full product kernels, while Matérn or mixed-product constructions provide additional control over smoothness and anisotropy.

## 5.2 TOY EXAMPLE

We first consider the one-dimensional example from Liu & Wang (2016). This is a Gaussian mixture with two components: $\pi = \frac{1}{3}\mathcal{N}(-2,1) + \frac{2}{3}\mathcal{N}(2,1)$. We use $M \in \{50, 200, 500\}$ particles and sample the initial particle set from $\mathcal{N}(0,1)$.

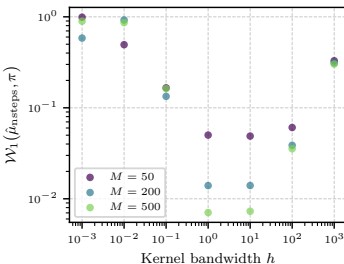

We run SVGD for $10^4$ steps with a step size of 1, using kernels of the form described above with $p = 1$ and different choices of (fixed) bandwidth $h$. As a measure of sample quality, we use the Wasserstein 1-distance $\mathcal{W}_1$, which we compute using an implementation of the explicit formula $\mathcal{W}_1(\mu,\nu) = \int_{\mathbb{R}} |F_\mu(x) - F_\nu(x)| \mathrm{d}x$ (see Panaretos & Zemel (2019)) and an exact sample of size $10^5$. The results of these experiments are shown in Figure 1, where we see that the algorithm performs well only for bandwidths within a certain range. The algorithm is highly sensitive to the choice of the parameter $h$ and therefore, a careful selection strategy is crucial.

Figure 1: Final Wasserstein 1-distances for one-dimensional example using SVGD with different fixed bandwidths $h$.

## 5.3 LINEAR INVERSE PROBLEM BASED ON ODE

The following example is adapted from (Weissmann et al., 2022, Example 2.1). We consider the one-dimensional differential equation

$$\begin{cases} -f''(s) + f(s) = u(s) & \text{for } s \in (0,1) \\ f(s) = 0 & \text{for } s \in \{0,1\} \end{cases} \quad (11)$$

and the associated inverse problem of recovering the right-hand side $u(\cdot) \in L^2([0,1])$ from discrete noisy observation points of the solution $f \in H^2([0,1]) \cap H_0^1([0,1])$. These observations are described by

$$y = \Phi(u) + \varepsilon \in \mathbb{R}^{N_{\text{obs}}},$$

where $\varepsilon \in \mathbb{R}^{N_{\text{obs}}}$ is observational noise and the forward operator $\Phi : L^2([0,1]) \to \mathbb{R}^{N_{\text{obs}}}$ is defined by $\mathcal{O} \circ H^{-1}$, with $H(f) = -f'' + f$ and $\mathcal{O}(f) = \big(f(s_1), \dots, f(s_{N_{\text{obs}}})\big)^\top \in \mathbb{R}^{N_{\text{obs}}}$ being the observation operator at $N_{\text{obs}}$ equidistant points $s_k = \frac{k}{N_{\text{obs}}}, k = 1, \dots, N_{\text{obs}}$.

For the Bayesian formulation of the inverse problem, we consider a Gaussian process (GP) prior for $u$ given by the truncated Karhunen-Loève (KL) expansion

$$u(\cdot, x) = Ax = \sum_{k=1}^{N_x} x_k \psi_i,$$

where $\psi_k(s) = \sqrt{2}\sin(\pi k s)$ and $x_k \sim \mathcal{N}(0, \lambda_k)$ independently with $\lambda_k = 50k^{-2}$. The resulting problem is to estimate the KL coefficients $x = (x_1, \dots, x_{N_x})^\top \in \mathbb{R}^{N_x}$ with prior given by $\mathcal{N}(0, \Gamma_0)$ with $\Gamma_0 = \text{diag}(\lambda_1, \dots, \lambda_{N_x})$. Assuming $\varepsilon \sim \mathcal{N}(0, \Gamma)$, this leads to the posterior density

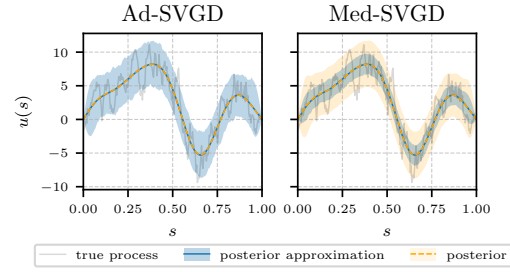

Figure 2: GP reconstruction for ODE-based inverse problem, showing mean and 90% confidence interval

$$\pi(x) \propto \exp\left(-\frac{1}{2}\|\Gamma^{-1/2}(y - \Phi Ax)\|^2 - \frac{1}{2}\|\Gamma_0^{-1/2}x\|^2\right), \quad x \in \mathbb{R}^{N_x}.$$

For the implementation, we replace $H$ by a numerical discretization operator for Equation (11) using a grid with mesh size $2^{-8}$ and consider the fully observed system (i.e. $N_{\text{obs}} = 2^8$). We use $N_x = 16$ terms for the KL expansion of $u$ and assume noise covariance $\Gamma = 10^{-3}\,\text{Id}_{N_{\text{obs}}}$. We construct reference observations $\bar{y} \in \mathbb{R}^{N_{\text{obs}}}$ by drawing $\bar{x} \sim \mathcal{N}(0, \Gamma_0)$ and setting $\bar{y} = \Phi A\bar{x}$.

We compare the performance of SVGD using the median heuristic (we call this Med-SVGD) with our Ad-SVGD for different choices of particle ensemble size $M \in \{50, 100, 200\}$. We use the kernels described in Section 5.1 with the choice $p = 1$. We run both algorithms for $4 \cdot 10^5$ iterations using the step size $10^{-3}$ for particle updates and, as suggested by Liu & Wang (2016), a variant of AdaGrad for adaptive step size control. In Ad-SVGD we use the step size $10^{-5}$ for the bandwidth updates and update the bandwidth only once for every 100 particle updates. With this update scheme, there was no significant runtime difference between Med-SVGD and Ad-SVGD.

Figure 2 shows the GP reconstruction for an exemplary seed. We observe that both methods are able to give a good approximation of the mean, but only Ad-SVGD correctly captures the posterior uncertainty. To further quantify the approximation quality, we use the Wasserstein 2-distance $\mathcal{W}_2\big(\mathcal{N}(\hat{\mu}, \hat{\Sigma}), \pi\big)$ between the posterior $\pi$ and the normal distribution $\mathcal{N}(\hat{\mu}, \hat{\Sigma})$, where $\hat{\mu}$ is the sample mean and $\hat{\Sigma}$ the sample covariance of the particle set (Panaretos & Zemel, 2019, Section 3.2). Since the target $\pi$ is a multivariate normal distribution, this has an explicit formula

$$\mathcal{W}_2\big(\mathcal{N}(\hat{\mu}, \hat{\Sigma}), \pi\big)^2 = \|\hat{\mu} - \mu_\pi\|^2 + \operatorname{trace}\left(\hat{\Sigma} + \Sigma_\pi - 2\big(\hat{\Sigma}^{1/2}\Sigma_\pi\hat{\Sigma}^{1/2}\big)^{1/2}\right),$$

where $\mu_\pi$ and $\Sigma_\pi$ are the mean and covariance of the posterior. We also compare the marginal variances of the final particle distribution with the posterior. Figure 3 shows the results of these experiments aggregated over 56 different random seeds (note that the posterior covariance does not actually depend on $\bar{y}$). Again, we observe that Ad-SVGD achieves better approximations of the posterior than Med-SVGD, which underestimates the uncertainty of the problem. Furthermore, in contrast to Med-SVGD, the approximation quality of Ad-SVGD improves as the number of particles increases beyond 50. Figure 4 shows the behavior of the bandwidths determined using Ad-SVGD. We observe that the component-wise bandwidths stabilize more quickly than the approximation error. Clear differences across the components are visible, with the final bandwidths being negatively correlated with the corresponding marginal variances.

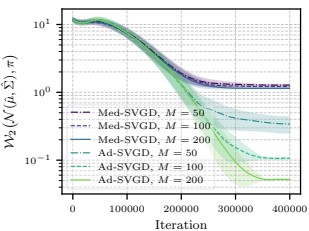
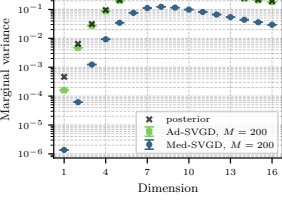

(a) Convergence of approximation error  (b) Marginal variances

Figure 3: Aggregated results (mean and 95% confidence interval over 56 random seeds) for ODE-based inverse problem using Med-SVGD and Ad-SVGD

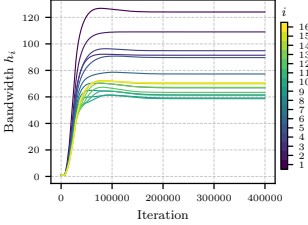
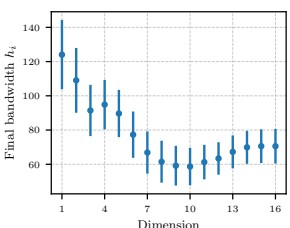

(a) Evolution of bandwidth per dimension (mean)  (b) Final bandwidths (mean and 95% conf. interval)

Figure 4: Behavior of bandwidth parameter for ODE-based inverse problem using Ad-SVGD, aggregated over 56 random seeds

## 5.4 BAYESIAN LOGISTIC REGRESSION

As in Liu & Wang (2016) and Liu et al. (2022), we consider a Bayesian logistic regression (BLR) model applied to the Covertype data set (Blackard, 1998). For the regression weights $w$, we assign

a Gaussian prior $w \mid \alpha \sim \mathcal{N}(0, \alpha^{-1})$ with $\alpha \sim \mathrm{Gamma}(1, 0.01)$ and we want to infer the posterior of $x = [w, \alpha]$. Following Liu et al. (2022), we use the No-U-Turn Sampler (NUTS) introduced by Hoffman & Gelman (2014) to generate a reference for evaluating the posterior approximation quality of Med-SVGD and Ad-SVGD. Again following Liu et al. (2022), we test the methods on subsets of the original data set of size 1000 and aggregate our results over 10 random draws. To evaluate the performance of the methods for the classification task, we compute the prediction accuracy over a test set achieved using the particle mean. As shown in Figure 5, Ad-SVGD outperforms Med-SVGD across almost all seeds, achieving a higher test accuracy while also being closer to the MCMC reference. To measure the accuracy of the posterior approximation, the same figure also shows the squared maximum mean discrepancy ($\mathrm{MMD}^2$) (Muandet et al., 2017, Section 3.5) to the MCMC samples. We observe a significant difference of how well the two methods are able to capture the posterior distribution. This is further confirmed by Figure 6, which shows the covariance matrices of the final particles compared to the MCMC reference, averaged over the 10 seeds (this visualization style is taken from Liu et al. (2022)). We observe that Med-SVGD severely underestimates the posterior uncertainty, while Ad-SVGD gives a better approximation. Additionally, the right plot in Figure 5 shows the evolution of the bandwidths for each component. We see that from an uninformed initialization (1 in every component), Ad-SVGD is able to recover suitable bandwidths, which differ from the initialization by several orders of magnitude.

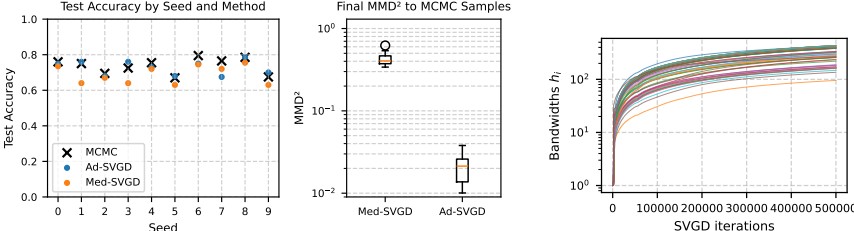

Figure 5: Approximation quality of SVGD particles for Bayesian logistic regression measured by prediction accuracy (left), $\mathrm{MMD}^2$ to reference samples (middle) and bandwidth evolution for BLR example (right)



Figure 6: Covariance matrices of final SVGD particle sets compared to MCMC samples (mean over 10 seeds)

## 6 LIMITATIONS AND OUTLOOK

The main limitation of our analysis is its reliance on Assumption 4. In our ongoing work, we examine when Assumption 4 is satisfied by the alternating gradient-ascent scheme in Algorithm 1 used in our experiments. Although we have no guarantees for Assumption 4 to be satisfied, our implementation led to promising empirical results.

Our considered analysis focused on the original dynamic of the SVGD, and it would be intriguing to combine the proposed adaptive kernel selection with recent variants such as sliced SVGD (Gong et al., 2021), Grassmann SVGD (Liu et al., 2022), or Stein transport (Nüsken, 2024).

## REPRODUCIBILITY STATEMENT

All parameters and procedures needed to reproduce our results are specified throughout the paper and appendix. We have submitted the complete codebase for running the experiments and generating the plots described in the main text.

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

## A    MORE NUMERICAL EXPERIMENTS

### A.1    GAUSSIAN MIXTURE MODELS

We revisit the example from Section 5.2. For the setup described there, Figure 8 shows histograms of the final particle distribution in comparison to the target density for $h \in \{0.001, 1, 1000\}$. We now turn to the comparison of Med-SVGD with Ad-SVGD. We consider different numbers of particles $M$ and compare the final particle distributions after $10^4$ iterations with step size 1 for the two methods.

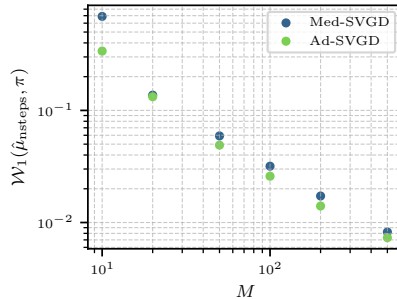

Figure 7: Approximation error for 1D Gaussian mixture.

Figure 7 shows the Wasserstein 1-distance between the final empirical distribution of the particles and the target distribution using Med-SVGD and Ad-SVGD with $M = 10, 20, 50, 100, 200, 500$. We observe that, as expected, the approximation quality improves with $N$ for both methods. Both methods achieve similar results, reaching a Wasserstein distance below 0.01 for $N = 500$.

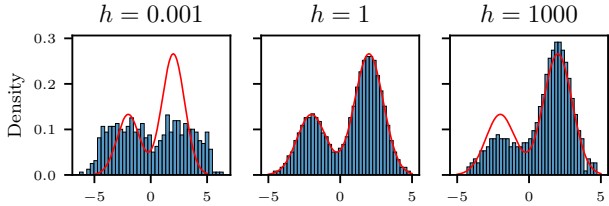

Figure 8: Histograms of final particle set generated using fixed bandwidths $h = 0.001, 1, 1000$; target density $\pi$ shown in red for comparison.

In the one-dimensional case, both methods are able to approximate the target distribution well. Our adaptive bandwidth selection strategy works well, but has no significant advantage over the commonly used median heuristic in this scenario.

### A.2    SCALING DIMENSION: MULTIVARIATE NORMAL DISTRIBUTION

Moving into higher dimensions, we now consider the Gaussian target distributions $\pi_d = \mathcal{N}(0, \Sigma_d)$ with $\Sigma_d = \text{diag}\left(1, \frac{1}{2}, \ldots, \frac{1}{d}\right)$ for $d \in \{2, \ldots, 8\}$. We used $\mathcal{N}(0, \frac{1}{d})^{\otimes d}$ as the initial particle distribution and ran the algorithms for $10^4$ iterations with step size 0.1. Whenever necessary for numerical stability, we used a smaller step size and adjusted the number of iterations accordingly.

We compare the approximation quality of the final set of particles generated using Med-SVGD and Ad-SVGD as the dimensionality of the problem increases. Because the Wasserstein 2-distance between two Gaussian distributions has an explicit formula (see Panaretos & Zemel (2019)), we use as a measure of sample quality the Wasserstein 2-distance between the target distribution $\pi_d$ and the Gaussian distribution $\mathcal{N}(\hat{\mu}, \hat{\Sigma})$, where $\hat{\mu}$ and $\hat{\Sigma}$ are the empirical mean and covariance matrix calculated from the set of particles (Panaretos & Zemel, 2019, Section 3.2). In accordance with the corresponding function of the Python Optimal Transport library Flamary et al. (2021), which we used for our calculations, we call this the *Bures Wasserstein distance*. Figure 9a shows the development of this sample quality measure, achieved using Med-SVGD and Ad-SVGD with $M = 50, 100, 200$, as the dimensionality of the problem increases. We see that Ad-SVGD significantly outperforms Med-SVGD for all values of $d$.

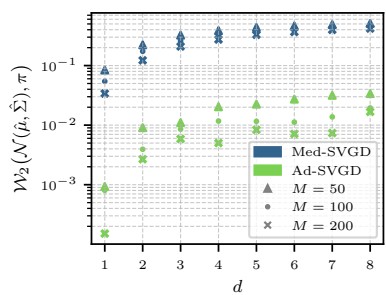
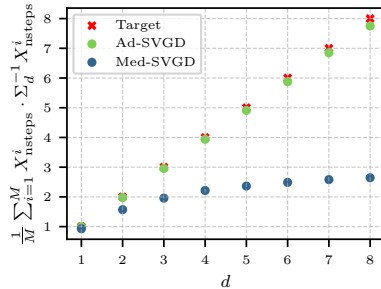

(a) Approximation error for $M \in \{50, 100, 200\}$.  (b) $\chi^2$-test statistic for $M = 200$.

Figure 9: Results for multivariate normal distributions of increasing dimension

Table 1: Marginal variances of final particle distribution for $d$-dimensional examples generated using Med-SVGD with $M = 200$; marginal variances of the target distribution $\pi_d$ shown for comparison.

| component | 1 | 2 | 3 | 4 | 5 | 6 | 7 | 8 |
|---|---|---|---|---|---|---|---|---|
| Target | 1.0000 | 0.2500 | 0.1111 | 0.0625 | 0.0400 | 0.0278 | 0.0204 | 0.0156 |
| $d = 1$ | 0.9285 | | | | | | | |
| $d = 2$ | 0.7921 | 0.1943 | | | | | | |
| $d = 3$ | 0.6803 | 0.1625 | 0.0697 | | | | | |
| $d = 4$ | 0.6089 | 0.1440 | 0.0593 | 0.0311 | | | | |
| $d = 5$ | 0.5532 | 0.1275 | 0.0526 | 0.0271 | 0.0157 | | | |
| $d = 6$ | 0.5190 | 0.1190 | 0.0481 | 0.0243 | 0.0140 | 0.0089 | | |
| $d = 7$ | 0.4900 | 0.1122 | 0.0449 | 0.0228 | 0.0131 | 0.0081 | 0.0052 | |
| $d = 8$ | 0.4753 | 0.1077 | 0.0430 | 0.0215 | 0.0122 | 0.0074 | 0.0047 | 0.0032 |

For the Bures Wasserstein distance, the smaller variances in the last components of our target distributions $\pi_d$ do not have a big impact; to see the impact of failing to correctly capture the uncertainty in these components, we consider the test statistic $X \cdot \Sigma_d^{-1} X$ which is $\chi^2$-distributed with $d$ degrees of freedom (i.e. it has expected value $d$) for $X \sim \mathcal{N}(0, \Sigma_d)$ (cf. (Sprungk et al., 2025, 27)). We calculate the mean of this statistic on the set of particles with $M = 200$ and compare it to the expected value $d$ in Figure 9b. Moving beyond the one-dimensional case, the test statistic for Med-SVGD deviates further and further from the true expected value as $d$ increases, while the statistic for Ad-SVGD stays relatively close to the true expectation. This shows the failure of Med-SVGD to correctly approximate higher-dimensional distributions, while Ad-SVGD is able to deal with those examples well.

We provide more details about the final particle distributions for $M = 200$ and different $d$ in Tables 1 and 2. They list, for each $d$, the marginal variances of the set of particles compared to the marginal variances of the target distribution, which are given in the first row. Table 1 shows the results for Med-SVGD, Table 2 shows the results for Ad-SVGD. We see that Ad-SVGD is able to achieve a good approximation of the target distribution in all components in terms of the marginal variances. The particles generated using Med-SVGD, on the other hand, significantly underestimate the uncertainty of the target distribution. This matches the observations already made in Figure 9b.

Lastly, we focus on the marginal particle distributions for $M = 200$ and $d = 8$. To ease the visualization, we normalized them by scaling each component of the particles with the inverse of the corresponding marginal standard deviation of the target distribution (i.e. we multiplied the $i$-th component with $i$). This turns each marginal distribution of $\pi_d$ into a standard normal distribution. Figures 10 and 11 show the histograms of these normalized marginal particles distributions for $d = 8$ generated using Med-SVGD and Ad-SVGD, respectively. A standard Gaussian density is shown in each plot for comparison. Again, we observe that Ad-SVGD is able to capture all marginal distributions well, while Med-SVGD underestimates the uncertainty of the target distribution. These observations are also visible in Figures 12 and 13, where the quantiles of the normalized marginal particle distributions are compared against the target quantiles (i.e. against a standard normal distribution).

Table 2: Marginal variances of final particle distribution for $d$-dimensional examples generated using Ad-SVGD with $M = 200$; marginal variances of the target distribution $\pi_d$ shown for comparison.

| component | 1 | 2 | 3 | 4 | 5 | 6 | 7 | 8 |
|---|---|---|---|---|---|---|---|---|
| Target | 1.0000 | 0.2500 | 0.1111 | 0.0625 | 0.0400 | 0.0278 | 0.0204 | 0.0156 |
| $d = 1$ | 0.9953 | | | | | | | |
| $d = 2$ | 0.9907 | 0.2472 | | | | | | |
| $d = 3$ | 0.9867 | 0.2459 | 0.1095 | | | | | |
| $d = 4$ | 0.9881 | 0.2467 | 0.1095 | 0.0610 | | | | |
| $d = 5$ | 0.9840 | 0.2433 | 0.1096 | 0.0616 | 0.0392 | | | |
| $d = 6$ | 0.9858 | 0.2459 | 0.1090 | 0.0611 | 0.0392 | 0.0269 | | |
| $d = 7$ | 0.9856 | 0.2463 | 0.1086 | 0.0613 | 0.0390 | 0.0269 | 0.0199 | |
| $d = 8$ | 0.9691 | 0.2409 | 0.1085 | 0.0611 | 0.0390 | 0.0268 | 0.0196 | 0.0150 |

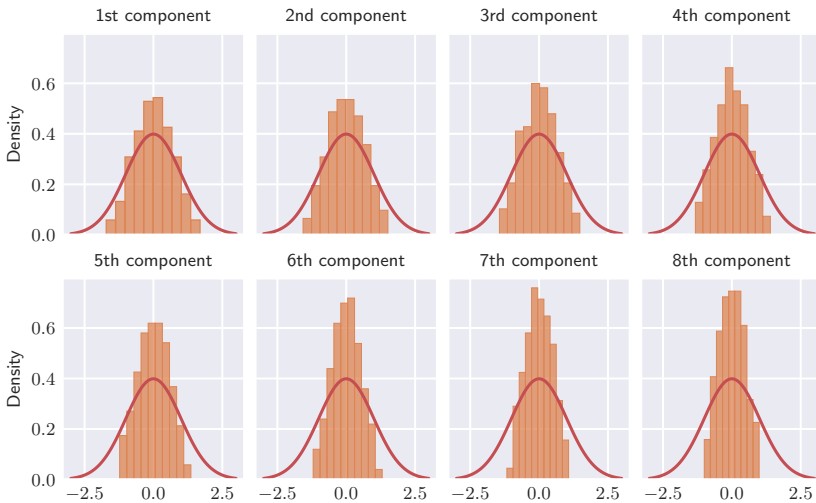

Figure 10: Histograms of single components of the final set of particles for eight-dimensional example generated using Med-SVGD with $M = 200$ and rescaled using marginal target variances; standard Gaussian density shown for comparison.

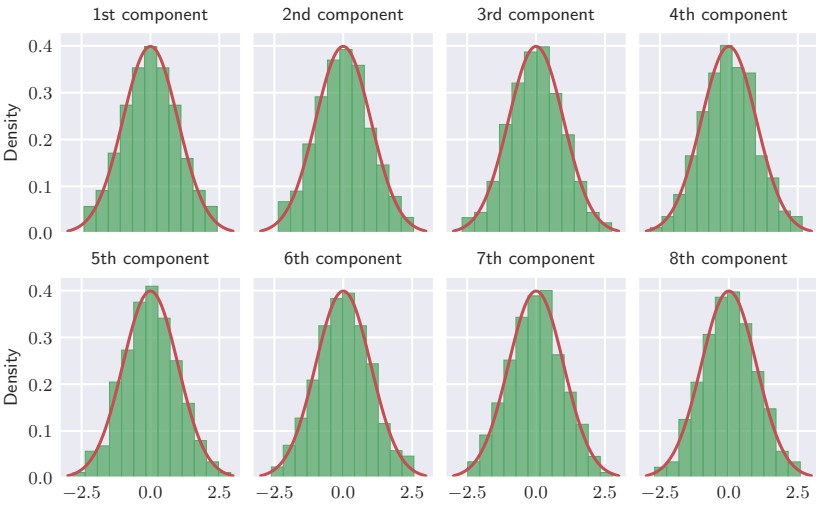

Figure 11: Histograms of single components of the final set of particles for eight-dimensional example generated using Ad-SVGD with $M = 200$ and rescaled using marginal target variances; standard Gaussian density shown for comparison.

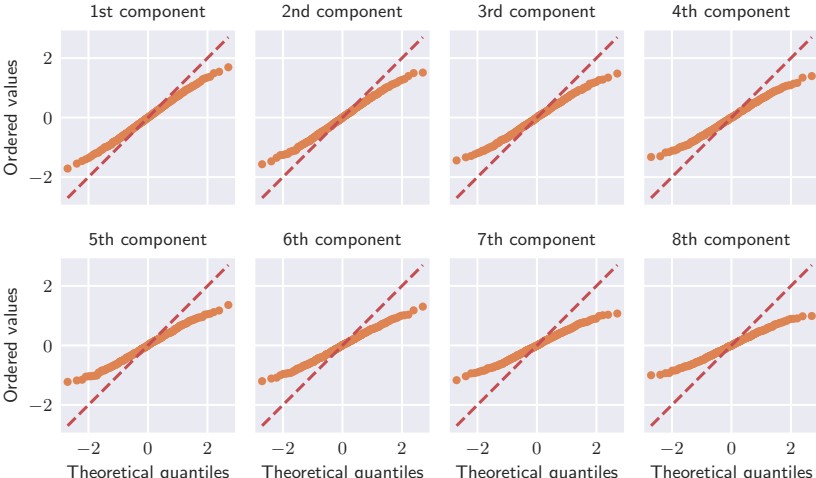

Figure 12: Q-Q plots comparing the marginals of the final particle distribution for eight-dimensional example generated using Med-SVGD with $M = 200$ and rescaled using marginal target variances with a standard normal distribution; line of slope 1 passing through the origin shown for comparison.

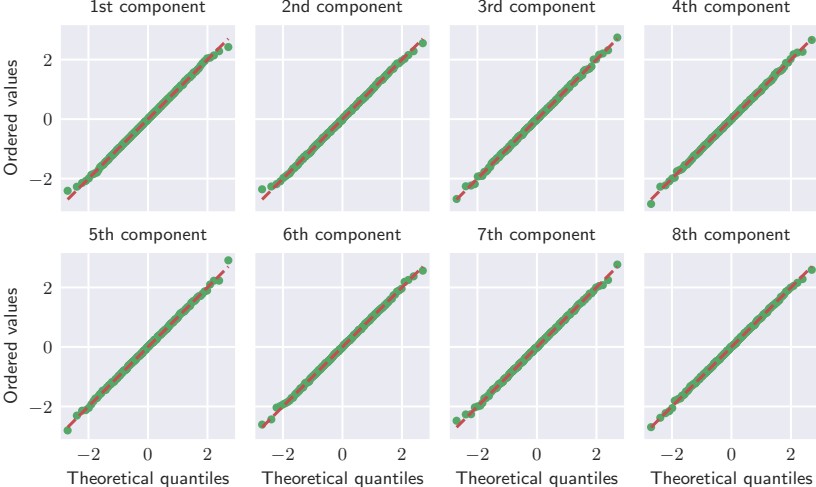

Figure 13: Q-Q plots comparing the marginals of the final particle distribution for eight-dimensional example generated using Ad-SVGD with $M = 200$ and rescaled using marginal target variances with a standard normal distribution; line of slope 1 passing through the origin shown for comparison.

### A.3 Gaussian process inference

We demonstrate the behavior of our Ad-SVGD for scaling dimensions in an inference task of a Gaussian process (GP) proposed in Reich & Weissmann (2021). We consider a GP on $[0, 1]$ represented by a truncated Karhunen-Loève (KL) expansion $u(s, x) = \sum_{k=1}^{N_x} x_k \psi_k(s)$ with basis functions $\psi_k(s) = \sqrt{2} \sin(k\pi s)$, where $x = (x_1, \ldots, x_{N_x})^\top$ is a vector of independent Gaussian random variables $x_k \sim \mathcal{N}(0, k^{-2})$. We observe the process at $N_y$ equispaced points in $[0, 1]$ and infer the coefficients $x_k$. For fixed $N_x$ and $N_y$, this corresponds to an inverse problem with forward model $Y = AX + \varepsilon$, where the $k$-th column of $A \in \mathbb{R}^{N_y \times N_x}$ is $(\psi_k(s_1), \ldots, \psi_k(s_{N_y}))^\top$, $s_i = \frac{i}{N_y}$ for $i = 1, \ldots, N_y$. The prior is $X \sim \mathcal{N}(0, \Sigma)$ with diagonal matrix $\Sigma$ with entries $k^{-2}$, $k = 1, \ldots, N_x$ and we assume independent Gaussian noise $\varepsilon \sim \mathcal{N}(0, I_{N_y})$. We construct reference observations $\bar{y} \in \mathbb{R}^{N_y}$ by drawing $\bar{x} \sim \mathcal{N}(0, \Sigma)$ and setting $\bar{y} = A\bar{x}$.

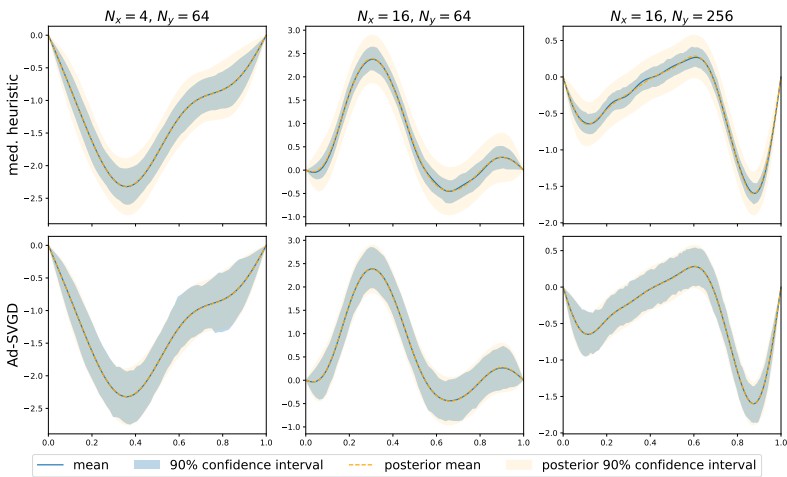

Figure 14: Estimated processes generated using the median heuristic and Ad-SVGD with $M = 100$ particles for different model configurations compared to posterior.

We use SVGD to sample from the resulting posterior $X \mid Y = \bar{y}$ and compare the performance of the median heuristic with our adaptive approach. For SVGD with the median heuristic, we use kernels of the form $k(x, y) = \exp(-\|x - y\|_1/h)$; for Ad-SVGD, we use product kernels $k(x, y) = \prod_{i=1}^{N_x} \exp(-|x_i - y_i|/h_i)$ with parameter $h = (h_1, \ldots, h_{N_x})$, i.e. we use a different bandwidth for each dimension. Following Liu & Wang (2016), we use a variant of Adagrad for step size control in the scenarios with $N_x = 16$.

Figure 14 shows the resulting processes in comparison to the posterior for the choices $N_x = 4$ and $N_y = 64$, $N_x = 16$ and $N_y = 64$ as well as $N_x = 16$ and $N_y = 256$. As the dimension increases, SVGD with the median heuristic underestimates the posterior variance, while Ad-SVGD is able to give a better approximation. The behavior is consistent across different numbers of observations of the Gaussian process. In Table 3, we quantify these results by comparing the trace of the covariance of the particle distributions generated by SVGD with the true posterior. SVGD with the median heuristic severely underestimates the uncertainty while Ad-SVGD is able to capture the variance more accurately.

| $N_x$ | 4 | 8 | 16 | 16 | 16 |
|---|---|---|---|---|---|
| $N_y$ | 64 | 64 | 64 | 128 | 256 |
| theoretical | 0.056 | 0.083 | 0.086 | 0.051 | 0.029 |
| med. heuristic | 0.026 | 0.023 | 0.022 | 0.012 | 0.006 |
| Ad-SVGD | 0.055 | 0.072 | 0.074 | 0.044 | 0.026 |

Table 3: Trace of covariance of final particle distribution ($M = 100$ particles) compared to theoretical posterior for different configurations, averaged over 25 runs.

## B  REFINED THEORETICAL ANALYSIS

We can further improve our convergence result under a so-called Stein log-Sobolev inequality, which relates the KL divergence to the KSD. Duncan et al. (2023) introduced this inequality and discussed conditions under which it might hold. Interestingly, in our adaptive setting of kernel selection, we can relax this condition by requiring only one kernel in our entire kernel class to satisfy the log-Sobolev inequality. More precisely, we make the following assumption.

**Assumption 5.** *Given a kernel class $\{k_\theta \mid \theta \in \Theta\}$, we assume that $\pi \in \mathcal{P}_1(\mathbb{R}^d)$ satisfies the generalized Stein log-Sobolev inequality with constant $\lambda > 0$:*

$$2\lambda \operatorname{KL}(\mu \parallel \pi) \leq \max_{\theta \in \Theta} \operatorname{KSD}_\theta^2(\mu | \pi)$$

*for all $\mu \in \mathcal{P}_1(\mathbb{R}^d)$.*

Using this property, we can quantify convergence in terms of the KL divergence. We emphasize that in this scenario we can give an error bound on $\mathcal{F}(\mu_n)$ even under an approximate condition $\varepsilon_n \leq \bar\varepsilon$ without requiring $\varepsilon_n \to 0$.

**Theorem 4.** *Suppose that Assumptions 1 to 3 and 5 are satisfied. For any $\alpha > 1$ and $\gamma > 0$ with*

$$\gamma \leq (\alpha - 1)\Big(\alpha B\big(1 + \|\nabla V(0)\| + L \int_{\mathbb{R}^d} \|x\| \, \mathrm{d}\pi(x) + L\sqrt{\frac{2\,\mathcal{F}(\mu_0)}{\lambda}}\big)\Big)^{-1},$$

*and*

$$\rho := 1 - 2\lambda c_\gamma \in (0, 1) \qquad where \qquad c_\gamma := \gamma(1 - \frac{\gamma B(\alpha^2 + L)}{2}),$$

*it holds that*

$$\mathcal{F}(\mu_{n+1}) \leq \rho \mathcal{F}(\mu_n) + c_\gamma \varepsilon_n\,,$$

*where $(\mu_n)_{n \in \mathbb{N}}$ is generated by (10) and $\varepsilon_n \geq \max_{\theta \in \Theta} \operatorname{KSD}_\theta^2(\mu_n \mid \pi) - \operatorname{KSD}_{\theta_n}^2(\mu_n \mid \pi)$. In particular, if*

1. *$\varepsilon_n \leq \bar\varepsilon$ for all $n \in \mathbb{N}$, then*

$$\mathcal{F}(\mu_n) \leq \rho^n\, \mathcal{F}(\mu_0) + \frac{c_\gamma\,\bar\varepsilon}{1 - \rho}$$

   *for all $n \in \mathbb{N}$.*

2. *$\varepsilon \leq c_\varepsilon q^n$ for some $q \in (0,1)$, $c_\varepsilon > 0$ and all $n \in \mathbb{N}$. Under $\rho \neq q$ we have*

$$\mathcal{F}(\mu_n) \in \mathcal{O}(\max\{\rho, q\}^n)$$

   *and under $\rho = q$ we have*

$$\mathcal{F}(\mu_n) \in \mathcal{O}(\tilde{q}^n)$$

   *for all $\rho < \tilde{q} < 1$.*

3. *$\varepsilon_n \leq \frac{c_\varepsilon}{(n+1)^p}$ for $c_\varepsilon, p > 0$ and all $n \in \mathbb{N}$, then*

$$\mathcal{F}(\mu_n) \leq \rho^n \mathcal{F}(\mu_0) + \frac{2^p\, c_\gamma\, c_\varepsilon}{n^p} \frac{1}{1 - \rho} + c_\gamma\, c_\varepsilon \frac{\rho^{\lfloor n/2 \rfloor}}{1 - \rho}$$

   *for all $n \in \mathbb{N}$.*

*Proof.* Similar to the proof of Theorem 3, under Assumption 5 and the conditions on $\alpha, \gamma$, we have

$$\mathcal{F}(\mu_{n+1}) \leq \mathcal{F}(\mu_n) - c_\gamma \max_{\theta \in \Theta} \operatorname{KSD}_\theta^2(\mu_n \mid \pi) + c_\gamma \varepsilon_n$$

$$\leq \rho \mathcal{F}(\mu_n) + c_\gamma \varepsilon_n\,,$$

where $c_\gamma = \gamma(1 - \frac{\gamma B(\alpha^2 + L)}{2}) > 0$. Iterating this recursion over $n \in \mathbb{N}$, we deduce the general error bound

$$\mathcal{F}(\mu_n) \leq \rho^n \mathcal{F}(\mu_0) + c_\gamma \sum_{k=0}^{n-1} \rho^{n-1-k} \varepsilon_k\,.$$

We consider the three different cases separately:

1. Under $\varepsilon_n \leq \bar{\varepsilon}$, the first claim follows by the computation of the geometric series

$$\sum_{k=0}^{n-1} \rho^{n-1-k} \leq \sum_{k=0}^{\infty} \rho^k = \frac{1}{1-\rho}\,.$$

2. Assuming that $\varepsilon_n \leq c_\varepsilon q^n$ for some $q \in (0,1)$ and $c_\varepsilon > 0$ we have

$$\mathcal{F}(\mu_n) \leq \rho^n \mathcal{F}(\mu_0) + c_\gamma c_\varepsilon \sum_{k=0}^{n-1} \rho^{n-1-k} q^k\,.$$

Suppose that $q > \rho$, then we have

$$\sum_{k=0}^{n-1} \rho^{n-1-k} q^k = \sum_{k=0}^{n-1} q^{n-1-k} \rho^k = q^{n-1} \sum_{k=0}^{n-1} (\rho/q)^k \leq q^{n-1} \frac{1}{1-\rho/q} = q^n \frac{1}{q-\rho}$$

where we have used the formula for geometric series. Similarly, in the case of $q < \rho$, we can directly bound

$$\sum_{k=0}^{n-1} \rho^{n-1-k} q^k = \rho^{n-1} \sum_{k=0}^{n-1} (q/\rho)^k \leq \rho^{n-1} \frac{1}{1-q/\rho} = \rho^n \frac{1}{(\rho-q)}\,.$$

Finally, in the case $q = \rho$, we simply bound $q^k < \tilde{q}^k$ for any $q < \tilde{q} < 1$ and deduce the claim line by line as before.

3. In the setting of $\varepsilon_n \leq \frac{c_\varepsilon}{(n+1)^p}$, we rewrite the upper bound on $\mathcal{F}(\mu_n)$ as

$$\mathcal{F}(\mu_n) \leq \rho^n \mathcal{F}(\mu_n) + c_\gamma \sum_{k=0}^{n-1} \rho^k \varepsilon_{n-1-k}$$

$$= \rho^n \mathcal{F}(\mu_0) + c_\gamma \Big( \sum_{k=0}^{\lfloor n/2 \rfloor} \rho^k \varepsilon_{n-1-k} + \sum_{k=\lfloor n/2 \rfloor+1}^{n-1} \rho^k \varepsilon_{n-1-k} \Big)$$

$$\leq \rho^n \mathcal{F}(\mu_0) + \frac{2^p c_\gamma c_\varepsilon}{n^p} \frac{1}{1-\rho} + c_\gamma c_\varepsilon \frac{\rho^{\lfloor n/2 \rfloor}}{1-\rho}\,.$$

Here, we have used $\varepsilon_{n-1-k} \leq \frac{c_\varepsilon}{(n-k)^p} \leq \frac{2^p c_\varepsilon}{n^p}$ for all $k \leq \lfloor n/2 \rfloor$ and $\varepsilon_k \leq c_\varepsilon$ for all $k \in \mathbb{N}$.

$\square$

