# OpenReview forum: "Adaptive kernel selection for Stein Variational Gradient Descent"
_ICLR.cc/2026/Conference — Submitted to ICLR 2026_

### Official Review · Reviewer_m3tA · 2025-10-30

**Soundness:** 3
**Presentation:** 3
**Contribution:** 2
**Rating:** 4
**Confidence:** 3

**Summary:**

This paper introduces an adaptive kernel selection approach, Ad-SVGD, for Stein Variational Gradient Descent (SVGD). The main idea is to dynamically adjust kernel parameters by maximizing the kernelized Stein discrepancy (KSD) during the SVGD process, such that the descent direction can better fit the data. The authors also provide a theoretical convergence analysis for the proposed method. The experimental results show the effectiveness of Ad-SVGD, which outperforms the standard SVGD with the median heuristic.

**Strengths:**

1. The idea of dynamically adjusting the kernel parameters using KSD is reasonable.

2. The theoretical convergence guarantee has been established.

**Weaknesses:**

1. From the theoretical perspective, it seems that some assumptions are too strong and not easy to check, e.g., Assumption 4, which depends on the behavior of the parameter to be optimized.

2. The experiments are not comprehensive enough. First, only a one-dimension toy example is considered in Section 5.2, and examples with higher-dimension should also be implemented. Second, only the original SVGD is used for comparison, while there are indeed lots of variants and improvements have been developed in recent years. Third, as a machine learning paper, tasks like classification and regression should be employed for evaluation, such as that used in the original SVGD paper and its followers.

3. The idea of adjusting or learning descent direction, with respect to certain criteria, for SVGD is not new, and related studies, e.g. [1][2], should be discussed or better compared with.

   [1] L. di Langosco et al. Neural Variational Gradient Descent. 4th Symposium on Advances in Approximate Bayesian Inference, 2022.

   [2] Q. Zhao et al. Stein variational gradient descent with learned direction. Information Sciences, 2023.

**Questions:**

The authors should address the issues mentioned in the "Weaknesses".

---

> ### Author Response · Authors · 2025-11-21
>
> Dear reviewer,
>
> we sincerely thank you for the time spent evaluating our work and for the constructive feedback. Below, we provide detailed responses to your concerns.
>
> ## Weakness 1 (Limited discussion of Assumption 4):
> We agree that Assumption 4 is difficult to verify from a theoretical standpoint.
> First, the finite multiple-kernel approach, where one uses a weighted sum of a fixed set of kernels, can be viewed as a special case of our proposed adaptive kernel selection framework. A key advantage of this parametrization is that the maximization of the KSD can be solved in closed form, eliminating the need for a gradient-ascent procedure. Consequently, in this setting our Theorem 3 applies without requiring Assumption 4.
>
> Second, we would like to emphasize that our numerical experiments were conducted without tuning to obtain highly accurate approximations of the maximal KSD at each iteration. In practice, we applied only a single gradient-ascent step every few iterations, yet the method still performed robustly. This empirical evidence suggests that Assumption 4 may be stronger than necessary and could potentially be relaxed.
>
> To clarify the role of Assumption 4 and investigate such relaxations, the revised manuscript now includes a refined iteration-complexity analysis under the Stein logarithmic Sobolev inequality, which relates the KSD to the KL divergence. Importantly, unlike vanilla SVGD, which requires this inequality to hold for a fixed kernel, our adaptive scheme only needs it to hold for at least one kernel in the considered kernel class.
>
> Moreover, in the refined analysis we provide an error bound on the KL divergence under a weaker assumption: instead of requiring the optimality gap in the KSD maximization to vanish, we assume only that this gap remains uniformly bounded. This leads to more flexible and broadly applicable theoretical guarantees.
> For further details, please refer to our general response to all reviewers, where we elaborate on the refined convergence analysis under the Stein logarithmic Sobolev condition.
>
> ## Weakness 2.1 (Only a one-dimension toy example):
>
> Thank you very much for raising this point, we acknowledge that the original manuscript did not sufficiently reference our additional experiments in the main text; this has now been corrected in the revised version. We would like to draw your attention to Appendices A2 and A3, where we present numerical toy examples illustrating the effect of scaling with dimension. In particular, the example in Appendix A3 clearly distinguishes between the dimensionality arising from numerical discretization and that stemming from random parametrization.
>
> ## Weakness 2.2 (Only original SVGD used for comparison):
>
> Thank you for raising this point. Our primary focus in this work is to investigate whether vanilla SVGD, without additional algorithmic tricks, can perform competitively when equipped with an effective, adaptively chosen kernel. For this reason, we restrict our comparisons to the original SVGD formulation, as it provides a clean baseline for isolating the impact of kernel selection itself.
>
> We agree that several SVGD variants (e.g., Sliced SVGD of Gong et al. (2021), Grassmann SVGD of Liu et al. (2022) and others) introduce additional structural components that may further improve performance. However, our adaptive kernel-selection strategy is orthogonal to these modifications and can, in principle, be combined with any of them. Our goal in this paper is to study the kernel-choice aspect in a controlled setting, rather than benchmark against methods whose benefits arise from fundamentally different mechanisms. We will clarify our research question in the contribution section.
>
> ## Weakness 2.3 (Limited exp validation):
>
> In the revised manuscript, we have added a new numerical experiment on the Bayesian logistic regression model used in the original SVGD paper by Liu \& Wang (2016). Please refer to our general response to all reviewers for further details.
>
> ## Weakness 3 (Related NN approaches should be discussed):
>
> Thank you for this comment. We were not aware of these references and will include them in our literature review.
>
> However, the methods referenced by the reviewer operate in a very different regime from ours: they rely on neural parameterizations that do not preserve the RKHS structure central to SVGD, and therefore lose the theoretical foundation on which SVGD is built. Our work addresses a fundamentally different research question. Namely, whether vanilla SVGD, without neural parametrizations or architectural heuristics, can achieve strong performance when equipped with principled, theoretically grounded kernel selection. The adaptive kernel selection mechanism we propose stays fully within the RKHS framework and maintains the connection to Stein operators, KSD minimization, and the associated convergence theory.

---

> > ### Author Response · Authors · 2025-11-21
> > **References**
> >
> > ## References:
> >
> > Gong et al. (2021): Sliced kernelized Stein discrepancy, International Conference on Learning Representations
> >
> > Liu \& Wang (2016): Stein variational gradient descent: A general purpose Bayesian inference algorithm, Advances in Neural Information Processing Systems
> >
> > Liu et al. (2022): Grassmann Stein variational gradient descent, Proceedings of The 25th International Conference on Artificial Intelligence and Statistics

---

### Official Review · Reviewer_JuCV · 2025-11-01

**Soundness:** 2
**Presentation:** 3
**Contribution:** 2
**Rating:** 2
**Confidence:** 3

**Summary:**

This paper aims to improve Stein Variational Gradient Descent (SVGD) by addressing one of its known limitations — the strong dependence on kernel choice. Standard SVGD typically uses a fixed kernel (such as RBF) with a manually chosen or heuristic bandwidth (e.g., the median heuristic). In practice, this choice heavily influences both stability and convergence, especially in high-dimensional or multimodal settings.

To mitigate this issue, the paper proposes Adaptive SVGD (Ad-SVGD), which dynamically updates kernel parameters (such as the bandwidth) during training. The kernel is optimized by maximizing the kernelized Stein discrepancy (KSD) at each iteration, based on the intuition that a larger KSD corresponds to faster reduction in the KL divergence between the particle distribution and the target posterior.

Experimental results indicate that Ad-SVGD can better maintain posterior variance and alleviate particle collapse compared to standard SVGD with a fixed bandwidth.

**Strengths:**

1. The dependence of SVGD on kernel choice is well-known. Adapting the kernel parameter through KSD maximization is an intuitive and well-motivated idea that could improve practical robustness.
2. The modification to SVGD is simple and lightweight, involving only an additional optimization step for the kernel parameter. It can be easily integrated into existing SVGD frameworks without altering the main update rule.

**Weaknesses:**

1. Most experiments are performed on toy or synthetic datasets with relatively low dimensionality. While results show qualitative improvements, more challenging or high-dimensional tests—such as Bayesian neural networks or large-scale posterior inference—would make the empirical claims more convincing.
2. The experiments are conducted mainly on small synthetic problems, so the practical effectiveness of the approach on larger or more complex tasks remains uncertain.
3. The paper does not discuss computational cost or scalability, even though the adaptive kernel adds extra optimization steps that could affect efficiency.
4. The comparison with other adaptive or multi-kernel SVGD variants is limited, making it difficult to clearly position this work among existing approaches.

**Questions:**

1. How sensitive is the method to the kernel learning rate and the number of inner optimization steps? Would overly aggressive kernel updates cause instability in particle dynamics?
2. If the kernel bandwidth is initialized poorly, can the adaptive process still recover a reasonable value, or does it tend to get stuck in suboptimal regions?
3. How does the adaptive kernel perform in higher-dimensional inference problems (e.g., 100–1000 dimensions)? Are there potential scalability issues related to the additional KSD computation?
4. It would strengthen the paper to include comparisons with related adaptive or multi-kernel approaches such as Multiple Kernel SVGD (Ai et al., 2023) and Matrix Kernel SVGD (Wang et al., 2019), to better position this work within the literature.
5. Since all current experiments are based on small or synthetic datasets (e.g., 1D Gaussian mixtures, linear ODE inverse problems, and small Gaussian process regression), it remains unclear whether the approach can handle complex or real-world posteriors, such as Bayesian neural networks.
6. The adaptive step introduces extra computation for matrix traces and kernel gradients, yet there is no discussion of runtime or complexity. Providing even a brief analysis would clarify the practical overhead.
7. The theoretical analysis assumes that the algorithm can approximately maximize KSD at each step, which may be difficult to guarantee in practice. Could the authors discuss how sensitive the theoretical results are to this assumption, and whether approximate optimization affects convergence?

---

> ### Author Response · Authors · 2025-11-21
>
> Dear reviewer,
>
> We sincerely thank you for the time spent evaluating our work and for the constructive feedback. Below, we provide detailed responses to your concerns.
>
> ## Weakness 1 \& 2 and Question 5 (Limited experiments):
>
> In the revised manuscript, we have added a Bayesian logistic regression experiment, a benchmark problem considered in Liu \& Wang (2016) and widely used in the SVGD literature. Please refer to our general response to all reviewers for further details.
>
> ## Weakness 3 and Question 6 (Missing discussion about computational cost):
>
> Thank you very much for raising this important point. First, we would like to emphasize that our adaptive scheme does not require any additional evaluations of the log-posterior density or its gradient. This is particularly relevant in applications where these evaluations dominate the computational cost.
> The additional overhead of Ad-SVGD arises solely from computing the kernel function and its derivatives (up to second order) during the KSD maximization step. The number of derivative evaluations necessary for each kernel update step scales quadratically with the number of particles, as the KSD is estimated as a double sum over the particles.
> In our numerical experiments, we did notice the impact of the additional computations for the adaptive kernel updates. However, our experiments showed that the effort can be kept under control by not updating the kernel at every step. To generate the results shown in the section about the new Bayesian logistic regression experiments, we only updated the kernel at every 500th step, which was still sufficient to achieve high-quality results. On our local machine, the average total runtime of Ad-SVGD was 1364 s, of which approximately 56.5 s were spent on the kernel update steps. This shows that in practical applications, the additional computational effort for the adaptive kernel updates is reasonable. It can be further alleviated by choosing a different kernel parametrization which does not use a kernel parameter for each dimension.
>
> ## Weakness 4 and Question 4 (Limited comparison with other adaptive SVGD variants):
>
> Thank you for raising this point. Our primary focus in this work is to investigate whether vanilla SVGD, without additional algorithmic modifications, can perform competitively when equipped with an effective and adaptively chosen kernel. For this reason, we restrict our comparisons to the original SVGD formulation, as it provides a clean baseline for isolating the impact of kernel selection itself.
>
> Specifically, the mentioned variant of SVGD using matrix valued kernels (Wang et al. (2019)) introduces additional geometric information based on Hessian and Fisher information matrices. This introduces an additional structural component that improves performance. We will add the paper to our literature review, thank you for making us aware of this reference. Moreover, we will clarify our research question in the contribution section.
>
> Regarding the multiple kernel approach of Ai et al. (2023), where one uses a weighted sum of a fixed set of kernels, we would like to emphasize that this approach can be seen as special case of our proposed adaptive kernel selection framework. A key advantage of this parametrization is that the maximization of the KSD can be solved in closed form, eliminating the need for a gradient-ascent procedure. Consequently, in this setting our Theorem 3 applies without requiring Assumption 4. However, our adaptive kernel selection scheme is more general, as it allows the kernel class to be parametrized over an uncountable set of kernel parameters. This flexibility is crucial when prior knowledge about how to hand-craft a small, effective set of kernels is unavailable. In such cases, the finite multiple-kernel approach may become impractical. For example, applying it to a product kernel in a 32-dimensional setting with 5 possible bandwidths per dimension would require $5^{32}$ kernels leading to severe computational overhead.

---

> > ### Author Response · Authors · 2025-11-21
> > **Continued Response**
> >
> > ## Question 1.1 (sensitivity to learning rate and number optimization steps):
> >
> > Thank you for this question. In our experiments, we intentionally did not devote significant effort to fine-tuning the kernel learning rate or the number of inner optimization steps, and we found the method to be reasonably robust across a wide range of settings. As with the step size in standard SVGD, the sensitivity of these hyperparameters is problem-dependent, and if needed, one could readily incorporate adaptive step-size strategies or more advanced optimizers to further stabilize or accelerate the inner kernel-optimization loop.
> >
> > ## Question 1.2 (overly aggressive kernel updates cause instability):
> >
> > In principle, instability can occur if an overly aggressive update leads to a very poor kernel choice. However, in our experiments we consistently observed that the evolution of the kernel parameters is typically well-behaved and often monotonic (see also Figures 4 and 5 of the revised manuscript). Furthermore, we experimented with setting the kernel bandwidth to its final value from the beginning (which corresponds to a very aggressive kernel update) and found that this did not cause instability in the particle dynamics.
> >
> > ## Question 2 (Can adaptive SVGD recover from suboptimal initial kernels?):
> >
> > We observed in some toy examples that extremely poor initializations can lead to unstable particle dynamics. Practical remedies include initializing with the median heuristic or adding a short warm-up phase during which the kernel is optimized while the particles remain fixed. In our ODE and Bayesian logistic regression experiments, we simply initialized all bandwidths to 1, and Ad-SVGD successfully recovered appropriate bandwidths, often differing by several orders of magnitude, which is now also reflected in the bandwidth-evolution plots (see Figures 4 and 5 of the revised manuscript).
> >
> > ## Question 3 (Performance in high dimensions and scalability issues)?
> >
> > We expect Ad-SVGD to remain more stable than median-heuristic SVGD in high dimensions; however, our current product-kernel parametrization reaches its limit in such settings because the number of parameters scales linearly with dimension. To address this, more scalable parametrizations can be incorporated into our framework, such as Matérn kernels, spectral kernels (based on truncated eigenfunction expansions, where singular values are learned while the basis is fixed) or low-rank kernel parametrizations that adapt the most relevant directions without requiring full product structures. These approaches avoid the exponential blow-up associated with high-dimensional product kernels and can be naturally integrated into our adaptive scheme. We have added a discussion about other possible parametrizations in Section 5.1 of the revised manuscript.
> >
> > ## Questions 4-6:
> >
> > See the responses to the weaknesses above.
> >
> > ## Question 7 (Limited discussion of Assumption 4):
> >
> > We agree that Assumption 4 is challenging to verify from a theoretical standpoint. However, we would like to emphasize that our numerical experiments were conducted without tuning to obtain highly accurate approximations of the maximal KSD at each iteration. In practice, we performed only a single gradient-ascent step every few iterations, yet the method still performed robustly. This empirical behavior suggests that Assumption 4 may be overly strict and could potentially be relaxed.
> >
> > To further clarify the role of Assumption 4 and explore possible relaxations, the revised manuscript now includes a refined iteration-complexity analysis under the Stein log-Sobolev inequality, which links the KSD to the KL divergence. In contrast to vanilla SVGD, where this condition must hold for a fixed kernel, our proposed adaptive scheme only requires that the inequality holds for at least one kernel within our kernel class. Additionally, we now provide an error bound on the KL divergence under a weaker assumption: instead of requiring the optimality gap in the KSD maximization to converge to zero, we assume only that this gap remains uniformly bounded.
> >
> > For more details, please refer to our general response to all reviewers, where we discuss the refined convergence analysis, as well as Appendix B of the revised manuscript.
> >
> > ## References:
> >
> > Ai et al. (2023): Stein Variational Gradient Descent with Multiple Kernels, Cognitive Computation
> >
> > Liu \& Wang (2016): Stein variational gradient descent: A general purpose Bayesian inference algorithm, Advances in Neural Information Processing Systems
> >
> > Wang et al. (2019): Stein variational gradient descent with
> > matrix-valued kernels, Advances in Neural Information Processing Systems

---

### Official Review · Reviewer_n3rK · 2025-11-05

**Soundness:** 3
**Presentation:** 3
**Contribution:** 3
**Rating:** 4
**Confidence:** 3

**Summary:**

The paper presents an adaptive kernel learning approach for Stein Variational Gradient Descent (SVGD) through the use of a parameterized family of kernels. The proposed method incorporates an additional optimization step to learn the optimal kernel parameters at each SVGD particle update. A convergence analysis for this adaptive setting is also provided, extending existing results for SVGD with fixed kernels. Experimental evaluations on synthetic datasets demonstrate that the proposed learning algorithm enhances the reliability and robustness of SVGD compared to a baseline SVGD with heuristic-based adaptive kernels.

**Strengths:**

The paper presents a meaningful contribution by introducing a parameterized kernel learning mechanism for Stein Variational Gradient Descent (SVGD), supported by theoretical analysis. The proposed approach adds an adaptive optimization step that learns kernel parameters during each SVGD update, thereby reducing dependence on manually selected kernels and heuristic tuning. The paper is clearly written and logically structured, with mathematical derivations and algorithmic steps explained in a way that is easy to follow. The experimental results, though limited to synthetic data, convincingly demonstrate that the method improves the reliability and adaptability of SVGD compared to heuristic-based adaptive kernel baselines.

**Weaknesses:**

While the proposed approach is promising, the experimental validation remains limited. The experiments are conducted only on synthetic datasets, without evaluation on real-world problems such as posterior estimation, one of the most prominent and successful applications of SVGD. This limitation makes it difficult to assess the practical usefulness and scalability of the proposed adaptive kernel learning method. Additionally, the chosen parameterized family of kernels appears relatively narrow in scope. A broader discussion or exploration of alternative parameterized kernel classes would strengthen the work and better demonstrate the generality and adaptability of the proposed framework.

From a theoretical perspective, the convergence analysis relies on several strong assumptions. It is unclear how these assumptions are satisfied or approximated in practical applications. In particular, since the authors acknowledge that Assumption 4 cannot be guaranteed, it would be valuable to include a more detailed discussion of how this limitation affects the theoretical results and whether any relaxation or empirical verification of this assumption is possible.

**Questions:**

Since the authors acknowledge that Assumption 4 cannot be guaranteed, how does this limitation affect the validity and interpretation of the convergence analysis results?

---

> ### Author Response · Authors · 2025-11-21
>
> Dear reviewer,
>
> We sincerely thank you for the time spent evaluating our work and for the constructive feedback. Below, we provide detailed responses to your concerns.
>
> ## Weakness 1 (Limited experimental validation):
>
> In the revised manuscript, we have added a new numerical experiment on the Bayesian logistic regression model used in the original SVGD paper by Liu \& Wang (2016). Please refer to our general response to all reviewers for further details.
>
> ## Weakness 2 (Limited discussion about kernel parametrization):
>
> Thank you very much for raising this point. We have added a detailed discussion about possible kernel parametrizations to our revised manuscript. In particular, we now discuss the relation to the multiple kernel approach from Ai et al. (2023) in more detail. Here one uses a weighted sum of a fixed set of kernels, which can be viewed as a special case of our proposed adaptive kernel selection framework. A key advantage of this parametrization is that the maximization of the KSD can be solved in closed form, eliminating the need for a gradient-ascent procedure. Consequently, in this setting our Theorem 3 applies without requiring Assumption 4. Moreover, our adaptive framework can accommodate a broad range of kernel parametrizations, including product kernels, Matérn kernels, spectral kernels (based on truncated eigenfunction expansions, where singular values are learned while the basis is fixed) or low-rank kernel parametrizations that adapt the most relevant directions without requiring full product structures. We added an additional discussion to our manuscript in Section 5.1.
>
> ## Weakness 3 and Question 1 (Limited discussion of Assumption 4; How does the limited verifiability of Assumption 4 affect the validity and interpretation of the convergence analysis?):
>
> We agree that Assumption 4 is challenging to verify from a theoretical standpoint. However, we would like to emphasize that our numerical experiments were conducted without tuning to obtain highly accurate approximations of the maximal KSD at each iteration. In practice, we performed only a single gradient-ascent step every few iterations, yet the method still performed robustly. This empirical behavior suggests that Assumption 4 may be overly strict and could potentially be relaxed.\\
> To further clarify the role of Assumption 4 and explore possible relaxations, the revised manuscript now includes a refined iteration-complexity analysis under the Stein log-Sobolev inequality, which links the KSD to the KL divergence. In contrast to vanilla SVGD, where this condition must hold for a fixed kernel, our proposed adaptive scheme only requires that the inequality holds for at least one kernel within our kernel class. Additionally, we now provide an error bound on the KL divergence under a weaker assumption: instead of requiring the optimality gap in the KSD maximization to converge to zero, we assume only that this gap remains uniformly bounded.
>
> For more details, please refer to our general response to all reviewers, where we discuss the refined convergence analysis, as well as Appendix B of the revised manuscript.
>
> ## References:
>
> Ai et al. (2023): Stein Variational Gradient Descent with Multiple Kernels, Cognitive Computation
>
> Liu \& Wang (2016): Stein variational gradient descent: A general purpose Bayesian inference algorithm, Advances in Neural Information Processing Systems

---

### Official Review · Reviewer_ree6 · 2025-11-08

**Soundness:** 3
**Presentation:** 3
**Contribution:** 2
**Rating:** 4
**Confidence:** 4

**Summary:**

This paper proposes a sampling algorithm, namely Adaptive Stein Variational Gradient Descent (**Ad-SVGD**), which is a modification of the standard SVGD by adaptively tuning the kernel parameters during sampling. The key idea is to maximize the kernelized Stein discrepancy (KSD) with respect to the kernel parameters at each iteration, thereby improving the rate of KL decrease and enhancing sample diversity.

The authors:

1. develop an adaptive kernel selection mechanism via gradient ascent on the KSD (**Algorithm 1**).

2. provide a convergence analysis extending prior KSD-based results (e.g., Salim et al., 2022) to adaptive kernels under Talagrand’s inequality (**Theorem 3**).

3. present empirical results on Gaussian mixtures, ODE-based inverse problems, showing that Ad-SVGD mitigates variance collapse and performs better than the median heuristic.

**Strengths:**

1. **Good motivation**: the choice of kernel is essential in improving the performance of SVGD. The Ad-SVGD provides a solution from the perspectively of maximizing the KL-decay from a set of parametrized kernels, which can provide insights to further study of the choice of kernels in SVGD.

2. **Clear Math Formulation**: the mathematical formulations of the Ad-SVGD and its convergence are sound and clear.

**Weaknesses:**

**The advantages of the proposed method Ad-SVGD is not studied completely**. In the introduction, the authors mentioned multiple advantages of using adapting kernels, including

(1) various kernels induce geometries can make SVGD converge in stronger metric

(2) larger KL decay to make the convergence faster.

However, none of these advantages are studied either theoretically or numerically in the paper: the current theory only show asymptotic convergence in maximum KSD within the parametrized kernel class, and it is not clear how the maximum KSD induces convergence in stronger metric and how the Ad-SVGD is more efficient than SVGD. In the numerical examples, the running times for Ad-SVGD and Med-SVGD are also not reported.

**Questions:**

1. What is the difference between using finite many kernels as in [1] and using parametrized kernels as in Ad-SVGD? What are the advantages of using parametrized kernels?

2. Based on Theorem 3 and results in [2], what is the iteration complexity comparison between Ad-SVGD and SVGD in the mean-field sense?

3. Typos: line 115, equation for $\mathcal{A}_\pi^k(x)$; line 155: proportional to KSD not squared KSD; in Assumption 4, the subindex $n+1$ should be $n$

---

> ### Author Response · Authors · 2025-11-21
>
> Dear reviewer,
>
> We sincerely thank you for the time spent evaluating our work and for the constructive feedback. Below, we provide detailed responses to your concerns.
>
> ## Weakness 1 (Does maximizing KSD induce convergence in a stronger metric?):
>
> Thank you very much for raising this question. In our revised manuscript, we provide a theoretical explanation of how our adaptive SVGD scheme leverages the maximal KSD to induce convergence in a stronger metric. In the theoretical analysis of SVGD, a popular assumption is the Stein log-Sobolev inequality, which relates the KSD to the KL divergence. In contrast to vanilla SVGD, where this condition must hold for a fixed kernel, our proposed adaptive scheme only requires that the inequality holds for at least one kernel within our kernel class. For more details, please refer to our general response to all reviewers, where we discuss the refined convergence analysis under the Stein log-Sobolev condition.
>
> ## Weakness 2 and Question 2 (Is Ad-SVGD more efficient than SVGD? What is the iteration complexity comparison between Ad-SVGD and SVGD in the mean-field sense?):
>
> The theoretical analysis in our original submission did not provide iteration-complexity results in the mean-field regime, and therefore did not allow a direct comparison between Ad-SVGD and vanilla SVGD. In the revised manuscript, we now include a detailed analysis of the iteration complexity of Ad-SVGD under the Stein log-Sobolev inequality, as mentioned above. When the maximization of the KSD can be computed in closed form, both SVGD and Ad-SVGD exhibit the same linear convergence rate. However, the convergence guarantees for Ad-SVGD rely on a more flexible Stein log-Sobolev inequality. In particular, it may be possible to verify a strictly better constant in Assumption 5 for Ad-SVGD than for vanilla SVGD. In such a case, Ad-SVGD would indeed achieve a faster linear convergence rate.
>
> When the maximization of the KSD cannot be solved analytically and instead must be approximated, Ad-SVGD may converge sublinearly. This represents a trade-off: the method may converge more slowly per iteration, but in exchange it enjoys convergence guarantees under a substantially weaker assumption.
>
> In our numerical experiments, we did notice the impact of the additional computations for the adaptive kernel updates. However, our experiments showed that the effort can be kept under control by not updating the kernel at every step. To generate the results shown in the section about the new Bayesian logistic regression experiments, we only updated the kernel at every 500th step, which was still sufficient to achieve high-quality results. On our local machine, the average total runtime of Ad-SVGD was 1364 s, of which approximately 56.5 s were spent on the kernel update steps. This shows that in practical applications, the additional computational effort for the adaptive kernel updates is reasonable. It can be further alleviated by choosing a different kernel parametrization which does not use a kernel parameter for each dimension.
>
> ## Question 1 (What is the difference between finite multiple kernels and our approach? What are advantages of using parameterized kernels?):
>
> Thank you for this question and for highlighting that the distinction was not sufficiently discussed in the original submission.
>
> First, the finite multiple-kernel approach, where one uses a weighted sum of a fixed set of kernels, can be viewed as a special case of our proposed adaptive kernel selection framework. A key advantage of this parametrization is that the maximization of the KSD can be solved in closed form, eliminating the need for a gradient-ascent procedure. Consequently, in this setting our Theorem 3 applies without requiring Assumption 4.
>
> Second, our adaptive kernel selection scheme is more general, as it allows the kernel class to be parametrized over an uncountable set of kernel parameters. This flexibility is crucial when prior knowledge about how to hand-craft a small, effective set of kernels is unavailable. In such cases, the finite multiple-kernel approach may become impractical. For example, applying it to a product kernel in a 32-dimensional setting with 5 possible bandwidths per dimension would require $5^{32}$ kernels leading to severe computational overhead.
>
> ## Question 2 (What is the iteration complexity comparison between Ad-SVGD and SVGD in the mean-field sense?):
>
> See Weakness 2.
>
> ## Question 3 (Typos):
>
> Thank you very much for your careful read. We have corrected these typos.

---

> ### Comment · Reviewer_ree6 · 2025-11-21
>
> I appreciate the authors' clarifications and the revision on the manuscript. In my opinion, the updated manuscript improves a lot in presentation: the newly added experiments help readers better understand the Ad-SVGD and its difference to vanilla SVGD; the extended analysis in Section B also reflects algorithmic advantages of Ad-SVGD analytically.
>
> A minor suggestion is to highlight the newly added contents, for the convenience of other reviewers and the ACs.
>
> Regarding the new result Theorem 4, I have a few follow-up questions:
>
> (1) Is it straightforward to derive a practical version of it - assuming kernel conditions (e.g. smoothness, boundness and kernel-LSI) over the class {$\theta_n$} that is used in the algorithm rather than the whole parameter class $\Theta$?  This kind of result should help to get rid of the Assumption 4 that we can't verify. It also helps to understand the implemented-version of Ad-SVGD - *only updating the kernel every 500 steps with one step of gradient ascent*: we can track the kernels that are actually used in the algorithm and see how they benefit the generation compared to the heuristic ones.
>
> (2) If we look at the kernel conditions, it looks like there is a trade-off between better kernel-LSI constant and larger generation step-size. When the set $\Theta$ gets larger,
>
>    (i) the kernel-LSI constant $\lambda$ increases, which implies more contraction at each step. Hence the algorithm accelerates;
>
>    (ii) the (regularity) boundness parameter $B$ increases, which implies smaller step-size to ensure step-wise contraction, Hence we need to run more iterations of the algorithm to achieve the same macroscopic time.
>
> I am wondering if this trade-off is an issue when we actually implement the algorithm: if we naively choose a rich set of kernels, we may have the advantage of better geometric property, but at the expense of choosing small-step size to ensure the stability of the algorithm.
>
> ---
>
> Overall, I think the revision has already improved the paper a lot, and I will definitely consider to increase my score.

---

> > ### Author Response · Authors · 2025-11-25
> >
> > Dear Reviewer,
> >
> > thank you very much for your positive feedback and follow-up questions. For convenience, we have highlighted the changes made during the discussion phase in blue colour. Regarding your follow-up questions:
> >
> > ### (1) Proving kernel-LSI along the iteration
> > This is an insightful suggestion. A potential approach would require refining the kernel-LSI so that it holds uniformly over the entire kernel class, with a kernel-dependent constant $\lambda(\theta)\ge0$. Establishing such a result would likely involve (i) an ascent condition on the KSD with respect to the kernel parameter requiring some regularity or smoothness of the mapping $\theta\mapsto KSD(\theta)$, together with (ii) a descent condition for the KL divergence induced by the particle update. Under these ingredients, one may indeed hope to show a property such as $\lim\inf_{n\in\mathbb N}\ \lambda(\theta_n)>0$, thereby ensuring that the inequality persists (or even strengthens) across the iterations. However, developing this refined analysis requires substantial additional work and goes beyond the scope of the current manuscript. We view this as a highly interesting direction for future research.
> >
> > ### (2) Trade-off: larger kernel-LSI constant vs. larger step size
> > Thank you very much for this sharp and insightful observation. Our new theory indeed reveals an inherent trade-off between the strength of the kernel-LSI constant and the admissible step size. On the one hand, a larger kernel-LSI constant yields a stronger contraction in the KL divergence. On the other hand, enlarging the kernel class tends to increase the constant $B$, which governs the smoothness and boundedness conditions on the kernel function, and in turn forces us to choose smaller step sizes to satisfy the descent condition (7). Consequently, expanding the kernel class has two opposing effects: improving the kernel-LSI constant while simultaneously tightening the step size constraint due to larger values of $B$.
> > While we did not experiment with different kernel classes in our numerical experiments, we did notice that for some problems, the stability of the algorithm was more sensitive to the learning rate. Considering the insight about this trade-off, this could indicate that choosing a smaller kernel class could be beneficial in some situations to achieve stability of the algorithm for larger step sizes.
> > This interplay could be leveraged to design more advanced adaptive strategies. For example, one might combine our framework with ideas from multilevel SVGD (Alsup et al., 2022), using a warm-up phase based on a smaller kernel class and gradually enlarging it as the particle approximation becomes more accurate. Developing a rigorous convergence theory for such an approach would, however, require additional analysis, since unlike in Alsup et al. (2022) the contraction rate and the approximation error would no longer decouple additively but interact through the kernel class itself.
> >
> > We will include a brief discussion of both points in the next revised version of our manuscript.
> >
> > #### References:
> > Alsup, T., Venturi, L. & Peherstorfer, B.. (2022). Multilevel Stein variational gradient descent with applications to Bayesian inverse problems. *Proceedings of the 2nd Mathematical and Scientific Machine Learning Conference*, in *Proceedings of Machine Learning Research*

---

### Author Response · Authors · 2025-11-21
**General Response**

We thank all four reviewers for their very detailed and constructive feedback and suggestions, which will help to significantly strengthen the presentation of the paper. We have prepared individual responses to all mentioned weaknesses and questions. Moreover, we first address two major points that were raised by several reviewers.

## Additional numerical experiment:

We have extended our initial set of synthetic experiments with an application of our adaptive SVGD to Bayesian logistic regression on the Covertype dataset, a benchmark already used in several other papers (e.g. Liu \& Wang (2016) and Liu et al. (2022)). This additional experiment confirms that the advantages over the median heuristic seen in our initial experiments can also be observed for real-data applications, namely that Ad-SVGD provides a better approximation of the posterior by avoiding variance collapse (for a detailed presentation of the results, see Section 5.4 of the revised manuscript).

## Refined theoretical analysis:

In Appendix B of the revised manuscript, we extend our original theoretical result, which established convergence of the maximal KSD without providing a convergence rate. We now present full iteration complexity bounds for the KL divergence under a Stein log-Sobolev inequality. This inequality, introduced in Duncan et al. (2023), provides a functional relationship between the KL divergence and the KSD. Crucially, in our adaptive kernel selection setting, we can relax this condition: Instead of requiring the inequality to hold for a fixed kernel (as in vanilla SVGD), we only require that it holds for at least one kernel in the chosen kernel class. We formalize this as a generalized Stein log-Sobolev assumption (Assumption 5 in the appendix).

The resulting bounds quantify precisely how the approximation error affects the rate:
- With uniformly bounded errors $\epsilon_n\le \bar\epsilon$, the KL divergence converges linearly up to a residual proportional to $\bar\epsilon$. Hence, our analysis now also covers the effect of inexact maximization of the KSD.
- With geometrically decaying errors, the KL divergence converges linearly.
- With polynomially decaying errors, we obtain a corresponding polynomial-rate upper bound.

## References:
Duncan et al. (2023): On the geometry of Stein variational gradient descent, Journal of Machine Learning Research

Liu \& Wang (2016): Stein variational gradient descent: A general purpose Bayesian inference algorithm, Advances in Neural Information Processing Systems

Liu et al. (2022): Grassmann Stein variational gradient descent, Proceedings of The 25th International Conference on Artificial Intelligence and Statistics

---

### Meta-Review · Area_Chair_EYzf · 2025-12-29

**Summary:**

While the paper presents a well-motivated idea and the revision substantially improved the theoretical clarity, the remaining concerns are primarily empirical and scope-related and remain unresolved at this stage. In particular, the paper still lacks convincing validation on large-scale, high-dimensional, and realistic inference problems that are central to modern SVGD applications. Moreover, systematic comparisons with a broader range of recent SVGD variants are missing, making it difficult to clearly assess the practical advantage of Ad-SVGD relative to the state of the art. Finally, although the authors weakened the theoretical assumptions, there remains a gap between the convergence guarantees and fully verifiable, implementation-level behavior of adaptive kernel updates. Taken together, these outstanding issues limit the paper’s impact and maturity for acceptance at this time.

**Reviewer Concerns:**

- Reviewer ree6 questioned whether the benefits of Ad-SVGD are convincingly demonstrated, particularly how KSD maximization yields convergence in stronger metrics (e.g., KL) and whether Ad-SVGD is more efficient than standard SVGD. These concerns were largely addressed through added Stein log-Sobolev–based analysis, iteration-complexity discussion, clarification of parameterized kernels, and runtime reporting, with remaining points framed as implementation-level extensions.

- Reviewer n3rK raised concerns about limited experimental validation beyond synthetic settings and reliance on strong, hard-to-verify assumptions, especially Assumption 4. These were partly addressed by adding a Bayesian logistic regression experiment, broadening the kernel discussion, and weakening the theoretical assumptions, though empirical coverage remains limited.

- Reviewer m3tA was concerned about strong theoretical assumptions, limited experiments (low-dimensional settings and vanilla SVGD comparisons), and insufficient discussion of related work. These concerns were partially addressed by relaxing assumptions, adding higher-dimensional and Bayesian logistic regression experiments, and clarifying relations to prior methods.

- Reviewer JuCV emphasized limited experimental scale, missing discussion of computational cost and scalability, restricted comparisons with adaptive SVGD variants, and reliance on near-maximal KSD assumptions. These issues were partly mitigated through added experiments, runtime analysis, clearer positioning, and weaker convergence guarantees, though large-scale validation remains lacking.

- The remaining concerns primarily relate to empirical scope rather than correctness: in particular, the lack of extensive validation on large-scale, high-dimensional real-world inference problems, the absence of systematic comparisons with a wider range of recent SVGD variants, and the gap between theoretical assumptions and fully verifiable implementation-level guarantees for adaptive kernel updates.

**Reviewer Scores:**

- Reviewer ree6 (ruchanged).
Reason: While the revision substantially improved clarity and addressed the reviewer’s main technical questions, the remaining issues were exploratory in nature and did not clearly justify a definitive upward revision of the original score.

- Reviewer n3rK (ruchanged).
Reason: Although the revision substantially improved the discussion and added experiments and weaker theoretical guarantees, the remaining limitations regarding experimental breadth and assumption verifiability prevent a clear upward revision beyond a borderline accept assessment.

Reviewer m3tA (unchagend)
- Likely unchanged (4), as the revision improves clarity and scope but does not fully resolve the experimental breadth concerns.


Reviewer JuCV (unchanged)
- No change , since the revision improves analysis and clarity but does not fully resolve the lack of high-dimensional, large-scale experiments.

---

### Decision · Program_Chairs · 2026-01-26

Reject